# COMPARING NOISY NEURAL POPULATION DYNAMICS USING OPTIMAL TRANSPORT DISTANCES

**Amin Nejatbakhsh**[1]**, Victor Geadah**[1,2]**, Alex H. Williams**[1,3] **& David Lipshutz**[1,4]

[1] Center for Computation Neuroscience, Flatiron Institute
[2] Applied and Computational Mathematics, Princeton University
[3] Center for Neural Science, New York University
[4] Department of Neuroscience, Baylor College of Medicine

anejatbakhsh@flatironinstitute.org

## ABSTRACT

Biological and artificial neural systems form high-dimensional neural representations that underpin their computational capabilities. Methods for quantifying geometric similarity in neural representations have become a popular tool for identifying computational principles that are potentially shared across neural systems. These methods generally assume that neural responses are deterministic and static. However, responses of biological systems, and some artificial systems, are noisy and dynamically unfold over time. Furthermore, these characteristics can have substantial influence on a system's computational capabilities. Here, we demonstrate that existing metrics can fail to capture key differences between neural systems with noisy dynamic responses. We then propose a metric for comparing the geometry of noisy neural trajectories, which can be derived as an optimal transport distance between Gaussian processes. We use the metric to compare models of neural responses in different regions of the motor system and to compare the dynamics of latent diffusion models for text-to-image synthesis.

## 1 INTRODUCTION

Biological and artificial neural systems represent their environments and internal states as patterns of neural activity, or "neural representations". In an effort to understand general principals governing these representations, a large body of work has sought to quantify the extent to which they are similar across systems (Klabunde et al., 2023). Guided by the intuition that the geometry of a systems' representation may be related to its function (Mikolov et al., 2013; Harvey et al., 2024), several measures of geometric (dis)similarity have been proposed and applied to data, including Representational Similarity Analysis (RSA, Kriegeskorte et al., 2008), Centered Kernel Alignment (CKA, Kornblith et al., 2019), canonical correlations analysis (CCA, Raghu et al., 2017), and Procrustes shape distance (Williams et al., 2021; Ding et al., 2021).

Most of these existing (dis)similarity measures assume that neural activities are deterministic and static. However, in many case, one or more of these assumptions does not hold. For example, the responses of sensory neurons to the same stimulus vary across repeated measurements (Goris et al., 2014), and the structure of these variations is thought to be critical to a system's perceptual capabilities (Averbeck et al., 2006). In addition, neural activities in brain regions such as motor cortex evolve according to complex dynamical motifs that correspond to current and future actions (Vyas et al., 2020). Similarly, responses of artificial neural systems can be noisy (e.g., variational autoencoders) or dynamic (e.g., recurrent neural networks) or both (e.g., diffusion-based generative processes).

Quantifying similarity in either the stochastic or dynamic aspects of neural systems has been independently addressed in the literature. Duong et al. (2023) proposed Stochastic Shape Distance (SSD), a stochastic extension to Procrustes shape distance (Williams et al., 2021) for quantifying differences in trial-to-trial noise across networks without addressing recurrent dynamics. Ostrow

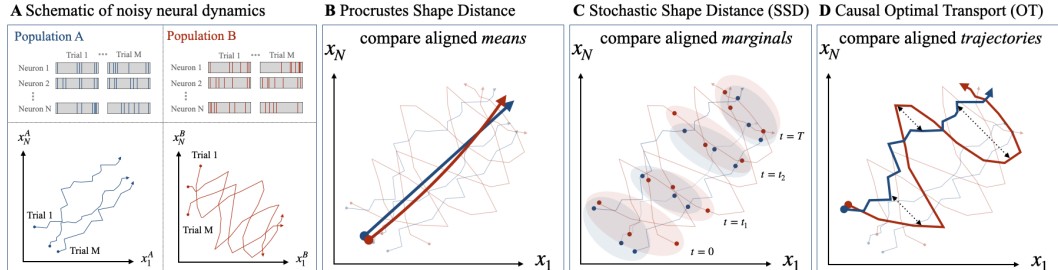

Figure 1: Metrics for comparing the shapes of noisy neural processes. **A)** Spike trains for two populations of neurons with noisy dynamic responses. Each trial can be represented as a trajectory in neural (firing rate) state space. (In the main text we account for cases that the populations have different numbers of neurons and trials.) **B)** Procrustes shape distance compares the aligned *mean trajectories* (thick curves represent the mean neural responses during a trial). **C)** Stochastic Shape Distance (SSD) compares the aligned *marginal statistics*. The faint ellipses represent the aligned marginal distributions at different times points. **D)** Our proposed metric (Causal OT distance) directly compares entire *trajectory statistics*, thus capturing across-time statistical dependencies.

et al. (2023) utilized Koopman operator theory to develop Dynamical Similarity Analysis (DSA) to compare recurrent flow fields without directly considering the effects of noise. However, on their own, these methods can fail to capture differences between noisy neural dynamics.

Here, we propose a novel metric between noisy and dynamic neural systems that captures differences in the systems' noisy neural trajectories (Fig. 1A). The metric is naturally viewed as an extension of Procrustes shape distance which compares average trajectories (Fig. 1B) and SSD which compares marginal statistics or noise correlations (Fig. 1C). Conceptually, our metric compares the statistics of entire trajectories (Fig. 1D). In particular, the metric can be derived as a *(bi-)causal optimal transport (OT) distance* (Lassalle, 2018; Backhoff et al., 2017) in which two systems are compared by computing the cost of "transporting" a set of trajectories generated by one system to match a set of trajectories generated by another system, and the mapping between trajectories must satisfy a temporal causality condition. The causality condition is especially relevant when comparing neural systems whose trajectories are close in space and yet the trajectories vary significantly in the amount of information that is available at different time points (see Sec. 4.1 for simple illustrative example). We apply our method to compare simplified models of motor systems (Sec. 4.2) and the dynamics of conditional latent diffusion models (Sec. 4.3).

In summary, our main contributions are:

- We demonstrate how existing metrics for comparing neural responses can fail to capture differences between neural population dynamics (Sec. 2).

- We propose a metric for comparing noisy neural population dynamics that is motivated by causal optimal transport distances between Gaussian processes (Sec. 3 and Appx. A).

- We provide an alternating minimization algorithm for computing the distance between two processes using their first- and second-order statistics (Appx. B). [1]

- We apply our method to compare models of neural responses in different regions of the motor system and to compare the dynamics of conditional latent diffusion models for text-to-image synthesis (Sec. 4).

## 2 SET UP AND EXISTING METHODS

Consider a neural system that consists of $N$ neurons whose activities (e.g., firing rates) at time $t \in \{1, \ldots, T\}$ are represented by an $N$-dimensional random vector $\mathbf{x}(t)$. The sequence $\mathbf{x} = \{\mathbf{x}(t)\}$ defines a random trajectory, or stochastic process, in $\mathbb{R}^N$ (Fig. 1A). Given two or more such neural systems, we'd like to develop meaningful measures of their (dis)similarity.

---

[1]Source code: https://github.com/amin-nejat/netrep.

## 2.1    PROCRUSTES SHAPE DISTANCE FOR COMPARING MEAN RESPONSES

Perhaps the simplest way to compare two dynamical neural systems living in different coordinate systems is to compute the Procrustes shape distance on their mean trajectories (Fig. 1B). For processes $\mathbf{x}$ and $\mathbf{y}$ with mean responses $\boldsymbol{m}_x(t) := \mathbb{E}[\mathbf{x}(t)]$ and $\boldsymbol{m}_y(t) := \mathbb{E}[\mathbf{y}(t)]$, the (squared) Procrustes shape distance is:

$$d_{\text{Procrustes}}^2(\mathbf{x}, \mathbf{y}) := \min_{\boldsymbol{Q} \in O(N)} \sum_{t=1}^{T} \|\boldsymbol{m}_x(t) - \boldsymbol{Q}\boldsymbol{m}_y(t)\|_2^2, \tag{1}$$

where $O(N)$ denotes the set of orthogonal matrices. This metric conceptualizes each process as a deterministic trajectory through neural state space and equates two processes if their mean responses are equal up to a rotation and/or reflection of neural state space. A useful property of equation 1 is that it defines a mathematical metric on neural processes. In particular, the function $d_{\text{Procrustes}}(\cdot, \cdot)$ is symmetric and satisfies the triangle inequality, which is helpful to establish theoretical guarantees for many statistical analyses such as neighborhood-based clustering and regression (Cover & Hart, 1967; Dasgupta & Long, 2005).

A common goal in analyzing neural systems is to distinguish between the impacts of recurrent interactions and feedforward input drive (Sauerbrei et al., 2020), so a useful measure should distinguish systems with different recurrent interactions. Galgali et al. (2023, Figure 1e) shows a simple but illuminating example for why this is challenging. They construct three different flow fields and adversarially tune time-varying inputs for each system so that the trial-average trajectories match. By construction, distances based on comparing mean trajectories (e.g., Procrustes shape distance) will fail to capture differences between the three systems.

It is easy, even in linear systems, to construct examples where the mean trajectories are insufficient to distinguish between candidate dynamical models. For example, consider the case that the process $\mathbf{x}$ satisfies a linear stochastic dynamical system of the form

$$\mathbf{x}(t) = \boldsymbol{A}(t)\mathbf{x}(t-1) + \boldsymbol{b}(t) + \boldsymbol{\Sigma}(t)\mathbf{w}(t), \tag{2}$$

where $\boldsymbol{A}(t)$ is an $N \times N$ dynamics matrix, $\boldsymbol{b}(t)$ is the $N$-dimensional input-drive, $\boldsymbol{\Sigma}(t)$ is an $N \times N$ matrix that determines the noise structure, and $\mathbf{w}(t)$ is an independent $N$-dimensional Gaussian random vector with zero mean and identity covariance matrix. Taking expectations on either side, we see that its average trajectory evolves according to the deterministic dynamical system

$$\boldsymbol{m}_x(t) = \boldsymbol{A}(t)\boldsymbol{m}_x(t-1) + \boldsymbol{b}(t). \tag{3}$$

From this dynamics equation, we see that the evolution of the mean trajectory depends on the *net* contributions from the (mean) recurrent interactions $\boldsymbol{A}(t)\boldsymbol{m}_x(t-1)$ and the input-drive $\boldsymbol{b}(t)$. Consequently, it is impossible to distinguish between recurrent dynamics and input-driven dynamics of a linear system of the form in equation 2 only using the mean trajectory. Galgali et al. (2023) showed that such differences can sometimes be accounted for by considering stochastic fluctuations about the mean trajectories, which suggests using metrics that explicitly compare stochastic neural responses.

## 2.2    STOCHASTIC SHAPE DISTANCE FOR COMPARING NOISY RESPONSES

Duong et al. (2023) introduced generalizations of Procrustes shape distance to account for stochastic neural responses. While their method was originally developed for networks with noisy static responses in different conditions, it can be directly applied to compare noisy dynamic responses by treating each time point as a different condition. Here we focus on SSDs that compare the first and second-order marginal statistics at each time point. Specifically, define the marginal covariances of a process $\mathbf{x}$ with mean trajectory $\boldsymbol{m}_x$ by

$$\boldsymbol{P}_x(t) := \mathbb{E}\left[(\mathbf{x}(t) - \boldsymbol{m}_x(t))(\mathbf{x}(t) - \boldsymbol{m}_x(t))^\top\right], \qquad t = 1, \dots, T. \tag{4}$$

The $\alpha$-*Stochastic Shape Distance ($\alpha$-SSD)* between processes $\mathbf{x}$ and $\mathbf{y}$ (Fig. 1C) is

$$d_{\alpha\text{-SSD}}^2(\mathbf{x}, \mathbf{y}) := \min_{\mathbf{Q} \in O(N)} \sum_{t=1}^{T} \left\{ (2-\alpha)\|\boldsymbol{m}_x(t) - \mathbf{Q}\boldsymbol{m}_y(t)\|^2 + \alpha\mathcal{B}^2\left(\boldsymbol{P}_x(t), \mathbf{Q}\boldsymbol{P}_y(t)\mathbf{Q}^\top\right) \right\}, \tag{5}$$

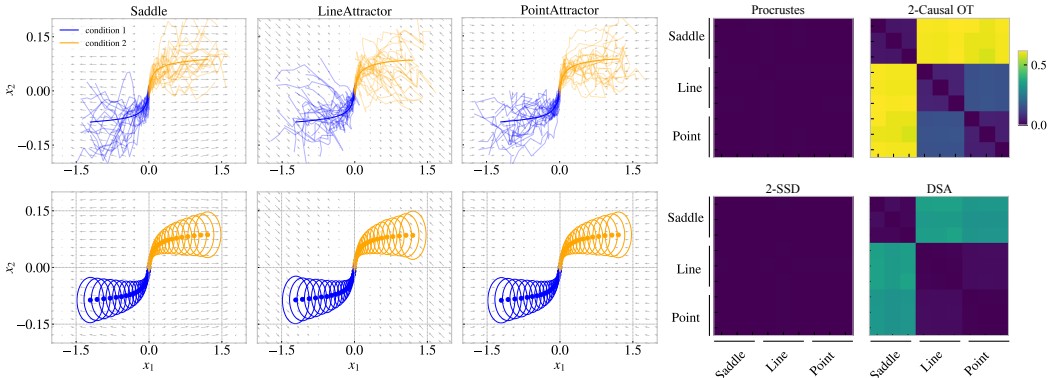

Figure 2: Demonstration that causal optimal transport (OT) distance disambiguates recurrent flow fields while distances based on the mean trajectories and the marginal noise correlations do not. On the left are simulations of three dynamical systems—a nonlinear dynamical system with two stable fixed points, a linear dynamical system with a line attractor and a linear dynamical system with a point attractor—with identical mean trajectories (top) and noise correlations (bottom) due to adversarially chosen input drives and input noise. On the right are the pairwise distances between the three systems.

where $O(N)$ is the set of orthogonal matrices, $\mathcal{B}(\cdot, \cdot)$ denotes the *Bures metric* between positive semidefinite matrices:

$$\mathcal{B}(\boldsymbol{A}, \boldsymbol{B}) = \min_{\boldsymbol{U} \in O(N)} \|\boldsymbol{A}^{1/2} - \boldsymbol{B}^{1/2} \boldsymbol{U}\|_F, \tag{6}$$

and $0 \leq \alpha \leq 2$ determines the relative weight placed on differences between the mean trajectories or the marginal covariances. When $\alpha = 0$, the distance reduces to the Procrustes shape distance between the mean trajectories $\boldsymbol{m}_x$ and $\boldsymbol{m}_y$, so $\alpha$-SSD can be viewed as a generalization that accounts for second-order fluctuations of the marginal responses. Furthermore, if $\alpha = 1$ and the marginal distributions are Gaussian, i.e., $\mathbf{x}(t) \sim \mathcal{N}(\boldsymbol{m}_x(t), \boldsymbol{P}_x(t))$ and $\mathbf{y}(t) \sim \mathcal{N}(\boldsymbol{m}_y(t), \boldsymbol{P}_y(t))$ for all $t = 1, \ldots, T$, then equation 5 is the sum of squared Wasserstein distances (with $p = 2$) between these marginal distributions (Peyré & Cuturi, 2019). This interpretation of SSD as an optimal transport distance between Gaussian distributions is relevant to the metric we develop in section 3 to compare full dynamical systems.

Lipshutz et al. (2024) showed that $\alpha$-SSD with $\alpha \in \{1, 2\}$ can succesfully differentiate systems with the same mean trajectories. To understand this, consider the special case that $\mathbf{x}$ satisfies equation 2. Then its marginal covariance matrix evolves according to the dynamics

$$\boldsymbol{P}_x(t) = \boldsymbol{A}(t) \boldsymbol{P}_x(t-1) \boldsymbol{A}(t)^\top + \boldsymbol{\Sigma}(t) \boldsymbol{\Sigma}(t)^\top. \tag{7}$$

From this equation, we see that the evolution of the covariance matrix $\boldsymbol{P}(t)$ does not depend on the input-drive $\boldsymbol{b}(t)$, so it can be used to compare the recurrent dynamics between two systems that share a common noise structure; that is, when $\boldsymbol{P}_x(0) = \boldsymbol{P}_y(0)$ and the input noise structure $\boldsymbol{\Sigma}(\cdot)$ is the same in both networks.

On the other hand, equation 7 also suggests that the marginal covariances alone cannot distinguish between contributions from the recurrent dynamics $\boldsymbol{A}(t)$ and contributions from the noise covariance $\boldsymbol{\Sigma}(t)$. We show this in Fig. 2 where we extend the example of Galgali et al. (2023) by adversarially tuning the input noise $\boldsymbol{\Sigma}(t)$ in addition to the input drive $\boldsymbol{b}(t)$ to achieve systems with different underlying recurrent dynamics, but the same mean and marginal covariance trajectories (see Appx. D for details). This demonstrates that only comparing marginal distributions is insufficient for comparing Gaussian processes of the form in equation 2. In particular, these metrics do not account for *across-time* statistical dependencies.

## 2.3 DYNAMICAL SIMILARITY ANALYSIS FOR COMPARING RECURRENT FLOW FIELDS

Ostrow et al. (2023) proposed DSA for comparing recurrent neural dynamics. Their method is based on Hankel Alternative View of Koopman (HAVOK) analysis (Brunton et al., 2017), which is

a data-driven method for obtaining linear representations of strongly nonlinear systems that leverages Koopman operator theory (Koopman, 1931) and implicitly accounts for across-time statistical dependencies. Essentially, DSA involves constructing a Hankel matrix by delay-embedding a system's observed trajectories and then fitting a reduced-rank regression model to obtain a linear estimate (i.e., a fixed dynamics matrix $\boldsymbol{A}$) for the evolution of the singular vectors of the Hankel matrix. Even though the approximation is linear, it can capture global nonlinear dynamics provided the delay is sufficiently long (Takens, 1981). However, this can also lead to difficulties in practice since the delay needs to be appropriately tuned to capture the nonlinear dynamics. The same procedure can be applied to another system and the linear approximations of the two systems' dynamics can be compared. Specifically, given two such dynamics matrices $\boldsymbol{A}_x$ and $\boldsymbol{A}_y$, the distance between $\mathbf{x}$ and $\mathbf{y}$ is computed as

$$d_{\mathrm{DSA}}(\mathbf{x}, \mathbf{y}) := \min_{\boldsymbol{Q} \in O(N)} \|\boldsymbol{A}_x - \boldsymbol{Q}\boldsymbol{A}_y\boldsymbol{Q}^\top\|_F,$$

where the similarity transform $\boldsymbol{A}_y \mapsto \boldsymbol{Q}\boldsymbol{A}_y\boldsymbol{Q}^\top$ is the vector-field analogue of the alignment step in Procrustes shape distance.

Ostrow et al. (2023) demonstrated that DSA can disambiguate dynamical systems with the same mean trajectories but different underlying recurrent dynamics. However, DSA is not explicitly designed to account for stochastic dynamics and does not distinguish systems with varying noise levels (Appx. F). In addition, DSA depends on 2 hyperparameters: the number of delays used when constructing the Hankel matrix and the choice of rank in the reduced-rank regression. When these hyperparameters are optimally chosen, DSA successfully differentiates the dynamical systems in the extended example with adversarially tuned input noise (Fig. 2); however, DSA can fail when the hyperparameters are not optimally chosen (Appx. G).[2]

## 3  CAUSAL OPTIMAL TRANSPORT DISTANCES

We define a metric on noisy dynamic neural processes that accounts for both stochastic and dynamic aspects of neural responses. The metric is motivated by Causal OT distances on stochastic processes (Lassalle, 2018; Backhoff et al., 2017) and can conceptually be interpreted as the minimal $L^2$-cost for transporting a random set of trajectories generated by process $\mathbf{x}$ to a random set of trajectories generated by process $\mathbf{y}$, where the mapping (or coupling) between trajectories must satisfy a causality (or adapted) property that can be parsed as "the past of $\mathbf{x}$ is independent of the future of $\mathbf{y}$ given the past of $\mathbf{y}$" and vice versa. More specifically, the metric can be derived as the minimal $L^2$-cost between coupled Gaussian processes $\hat{\mathbf{x}}$ and $\hat{\mathbf{y}}$ that are driven by the same white noise process $\mathbf{w}$ and whose first two moments respectively match the first two moments of $\mathbf{x}$ and $\mathbf{y}$. The coupling is flexible in the sense that at each time $t$, the white noise vector $\mathbf{w}(t)$ driving $\mathbf{y}(t)$ can be *spatially* rotated by an orthogonal matrix $\boldsymbol{R}_t \in O(N)$ because this does not change its distribution and it preserves the temporal ordering of the white noise process. A detailed derivation of the distance is provided in Appx. A.

### 3.1  DEFINITION OF CAUSAL OPTIMAL TRANSPORT DISTANCE

Given a $N$-dimensional process $\mathbf{x} = \{\mathbf{x}(t)\}$ defined for $t = 1, \ldots, T$, let $\boldsymbol{C}_x$ be the $NT \times NT$ covariance matrix following a $T \times T$ block structure:

$$\boldsymbol{C}_x = \begin{bmatrix} \boldsymbol{P}_x(1,1) & \ldots & \boldsymbol{P}_x(1,T) \\ \vdots & \ddots & \vdots \\ \boldsymbol{P}_x(T,1) & \ldots & \boldsymbol{P}_x(T,T) \end{bmatrix}, \quad \boldsymbol{P}_x(s,t) := \mathbb{E}\left[(\mathbf{x}(s) - \boldsymbol{m}_x(s))(\mathbf{x}(t) - \boldsymbol{m}_x(t))^\top\right],$$

where the blocks $\boldsymbol{P}_x(s,t)$ are $N \times N$ matrices encoding the across-time covariances. The *$\alpha$-Causal OT* distance between two processes $\mathbf{x}$ and $\mathbf{y}$ is defined to be

$$d_{\alpha\text{-causal}}(\mathbf{x}, \mathbf{y}) :=$$

$$\min_{\boldsymbol{Q} \in O(N)} \left\{ \sum_{t=1}^{T} (2 - \alpha)\|\boldsymbol{m}_x(t) - \boldsymbol{Q}\boldsymbol{m}_y(t)\|^2 + \alpha \mathcal{AB}_{N,T}^2(\boldsymbol{C}_x, (\boldsymbol{I}_T \otimes \boldsymbol{Q})\boldsymbol{C}_y(\boldsymbol{I}_T \otimes \boldsymbol{Q}^\top)) \right\}, \quad (8)$$

---

[2] We implemented DSA using the code provided in the GitHub respository https://github.com/mitchellostrow/DSA.

where, as with $\alpha$-SSD, $0 \leq \alpha \leq 2$ determines the relative weight place on differences between the mean trajectories or the covariances, $\mathcal{AB}_{N,T}(\cdot, \cdot)$ is the *adapted Bures distance* on $NT \times NT$ positive semidefinite matrices defined by

$$\mathcal{AB}_{N,T}(\boldsymbol{A}, \boldsymbol{B}) := \min_{\boldsymbol{R}_1, \ldots, \boldsymbol{R}_T \in O(N)} \|\boldsymbol{L}_A - \boldsymbol{L}_B \text{diag}(\boldsymbol{R}_1, \ldots, \boldsymbol{R}_T)\|_F, \tag{9}$$

and $\boldsymbol{A} = \boldsymbol{L}_A \boldsymbol{L}_A^\top$ and $\boldsymbol{B} = \boldsymbol{L}_B \boldsymbol{L}_B^\top$ are the lower triangular Cholesky decompositions of $NT \times NT$ positive semidefinite matrices $\boldsymbol{A}$ and $\boldsymbol{B}$. Note that the Cholesky decomposition is unique for positive definite matrices. For a matrix with rank $R < N$, we take $\boldsymbol{L}$ to be the unique lower-triangular matrix with exactly $R$ positive diagonal elements and $N - R$ columns containing all zeros (Gentle, 2012).

## 3.2 PROPERTIES AND COMPARISON WITH EXISTING METRICS

Causal OT distance is naturally interpreted as an extension of Procrustes shape distance and SSD. When the across-time correlations are zero for both **x** and **y**, then $\boldsymbol{C}_x$ and $\boldsymbol{C}_y$ are block-diagonal and $\alpha$-Causal OT distance coincides with $\alpha$-SSD defined in equation 5. Therefore, Causal OT distance can be viewed as an extension of SSD that accounts for across-time correlations of stochastic processes. When applied to the dynamical systems with adversarially tuned input drives and input noise, we find that Causal OT can disambiguate the different dynamical systems that Procrustes shape distance and SSD cannot differentiate (Fig. 2).

It is also worth comparing Causal OT distance with Wasserstein (with $p = 2$) distance between Gaussian processes (Mallasto & Feragen, 2017), which treats the processes as Gaussian vectors without any notion of temporal ordering. The difference between the Wasserstein distance between two Gaussian process and equation 8 is that the adapted Bures distance $\mathcal{AB}_{N,T}(\cdot, \cdot)$ is replaced with the Bures distance $\mathcal{B}(\cdot, \cdot)$ between the covariance matrices:

$$\mathcal{B}(\boldsymbol{A}, \boldsymbol{B}) = \min_{\boldsymbol{U} \in O(NT)} \|\boldsymbol{A}^{1/2} - \boldsymbol{B}^{1/2}\boldsymbol{U}\|_F = \min_{\boldsymbol{U} \in O(NT)} \|\boldsymbol{L}_A - \boldsymbol{L}_B \boldsymbol{U}\|_F. \tag{10}$$

In the definition of Bures distance (equation 10), the minimization is over all orthogonal matrices $\boldsymbol{U} \in O(NT)$, whereas in the definition of adapted Bures distance (equation 9), the minimization is over the subset of block diagonal matrices $\{\text{diag}(\boldsymbol{R}_1, \ldots, \boldsymbol{R}_T), \boldsymbol{R}_1, \ldots, \boldsymbol{R}_T \in O(N)\}$. Therefore, the adapted Bures distance is always larger than the Bures distances: $\mathcal{AB}(\boldsymbol{A}, \boldsymbol{B}) \geq \mathcal{B}(\boldsymbol{A}, \boldsymbol{B})$. The minimizations over rotations in equation 9 and equation 10 essentially amount to differences in the allowed couplings. In the definition of adapted Bures distance (equation 9), the rotations are restricted to $N \times N$ *spatial* rotations, which implies that the couplings preserve a temporal ordering. However, in the definition of Bures distance (equation 10), there are no restrictions on the rotations, so the couplings can change the temporal ordering. This is important because two stochastic processes may have sample trajectories that are close (so Wasserstein distance is small) and yet the processes are very different in terms of how much information is available early in the trajectory. In section 4.1, we provide a tractable example that illustrates the effect of preserving temporal causality when comparing two processes.

## 4 EXPERIMENTS

### 4.1 A SCALAR EXAMPLE

We first consider an analytically tractable scalar example that highlights the temporal causality property of Causal OT, which is essentially a Gaussian process version of (Backhoff et al., 2022, figure 1). Consider two mean-zero scalar Gaussian processes $x = \{x(t)\}$ and $y = \{y(t)\}$ defined for $t = 0, 1, 2$ (e.g., firing rates of two neurons over three time steps) that both start at zero and end with distribution $\mathcal{N}(0, \sigma^2)$ for some $\sigma > 0$. The difference between them is that the first step of $x$ is small stochastic and the second step of $x$ is deterministic, whereas the first step of $y$ is deterministic and the second step of $y$ is stochastic (Fig. 3).

Specifically, for some $\epsilon > 0$ small, let

$$x(1) \sim \mathcal{N}(0, \epsilon^2), \qquad\qquad x(2) = \frac{\sigma}{\epsilon} x(1)$$

$$y(1) = 0, \qquad\qquad y(2) \sim \mathcal{N}(0, \sigma^2).$$

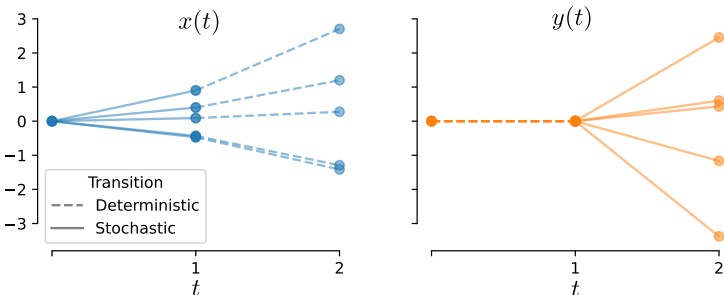

Figure 3: Five samples of two scalar processes over two time steps. The two processes have the same marginal distributions at time $t = 0$ and $t = 2$. The difference between the process is whether the first or second step is deterministic or stochastic. In this example, $\sigma = 1.5$ and $\epsilon = 0.5$.

When $\epsilon > 0$ is small, the sample trajectories of these processes are very close (Fig. 3). However, $x$ and $y$ are very different as stochastic processes: if $x(1)$ is known, then we have full information about $x(2)$, whereas if $y(1)$ is known, then we have no additional information about $y(2)$. In the context of a neural system, the first process could represent a neuron whose activity is ramping in time, whereas the second process could represent a neuron that is silent until the final time step.

The difference between the stochastic processes is captured by Causal OT, but not by Procrustes distance, SSD or Wasserstein. In particular, the distances are (see Appx. C for details):

$$d_{\text{Procrustes}}(x, y) = 0, \qquad\qquad d_{\text{1-SSD}}(x, y) = \epsilon$$

$$d_{\text{Wasserstein}}(x, y) = \epsilon, \qquad\qquad d_{\text{1-causal}}(x, y) = \sqrt{\sigma^2 + \epsilon^2}.$$

Procrustes distance cannot distinguish the two systems. Furthermore, as $\epsilon \to 0$, both $d_{\text{1-SSD}}(x, y)$ and $d_{\text{Wasserstein}}(x, y)$ converge to zero, whereas $d_{\text{1-causal}}(x, y)$ converges to $\sigma$. In particular, Causal OT captures differences in the across-time statistical structure of the stochastic processes that is not captured by SSD or Wasserstein.

## 4.2 Synthetic experiments modeling motor systems

We apply Causal OT distance to compare simple models of neural activity in motor systems. We take inspiration from experimental measurements of motor cortical dynamics in which preparatory and movement-related neural activity evolve into orthogonal subspaces (Kaufman et al., 2014; Churchland & Shenoy, 2024), Fig. 4A. This effectively creates a gating mechanism by which the neural activities can organize future movements without being "read out" into immediate motor actions. Here, we consider a simple set up comparing two "systems"—one system includes preparatory activity and motor activity that evolve in orthogonal subspaces and the other system only includes the motor activity. We show that Causal OT distance can differentiate these two systems even when preparatory activity is small (which has been observed experimentally, Churchland et al., 2012; Kaufman et al., 2016), while SSD cannot.

The first system is a two-dimensional process $\mathbf{x} = \{\mathbf{x}(t) = (x_1(t), x_2(t))\}$ that is a simple model of motor cortex dynamics during a reaching task (Churchland et al., 2012; Kaufman et al., 2014). The process is governed by the following time inhomogeneous linear system (explained below)

$$
\begin{array}{lll}
\text{Preparatory dynamics} & \mathbf{x}(t) = \mathbf{x}(t-1) + u\boldsymbol{e}_1 + \mathbf{w}(t) & \text{(11a)} \\
\text{Rotational dynamics} & \mathbf{x}(t) = \boldsymbol{M}\mathbf{x}(t-1) + \mathbf{w}(t) & \text{(11b)}
\end{array}
$$

where $\boldsymbol{e}_1, \boldsymbol{e}_2$ are the standard basis vectors in $\mathbb{R}^2$ and $\mathbf{w} = \{\mathbf{w}(t)\}$ is a two-dimensional additive Gaussian noise process (note that $\boldsymbol{e}_1$ is aligned with the vertical axis and $\boldsymbol{e}_2$ is aligned with the horizontal axis in Fig. 4A). Over the first $T_p \geq 1$ time-steps, the system evolves along the *preparatory dimension* (also referred to as the output-null direction in the literature) according to equation 11a, where $u$ takes values between $[-1, 1]$ that determines the future output of the system. This is followed by $T_r \geq 1$ time-steps of rotational dynamics, equation 11b, where $\boldsymbol{M}$ is the rotation matrix

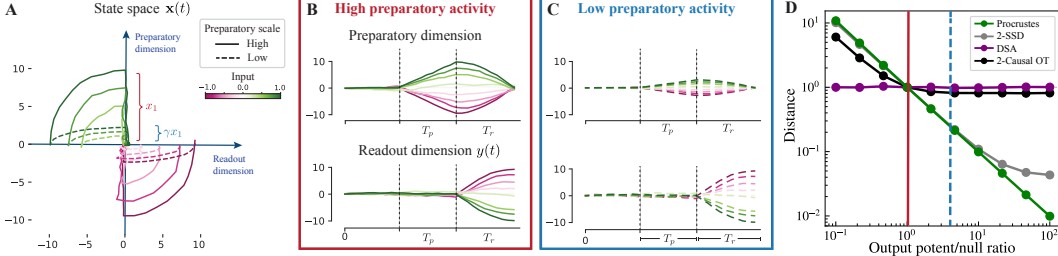

Figure 4: Motor cortex preparatory activity in the nullspace. **A)** State space activity $\mathbf{x}(t)$, accumulating inputs along preparatory dimension, before entering rotational dynamics with non-zero projection onto the readout dimension, showing different trajectories for different input scales (see colorbar). We show two different scales of preparatory activity in solid and dashed lines. **B)** Projections of state-space activity along the preparatory and readout dimensions. The readout $y(t)$ is a noisy version of the bottom row. **C)** Same as in **B** for the low preparatory activity scale, with scaling $\gamma = 0.3$. **D)** Distances as a function of the relative scale of the readout axis with respect to preparatory, normalized so that each distance is 1 when the output potent/null ratio is 1 (Procrustes shape distance and SSD overlap when the output potent/null ratio is between $10^{-1}$ and $10^{1}$).

defined by

$$\boldsymbol{M} := \begin{bmatrix} \cos(\theta) & -\sin(\theta) \\ \sin(\theta) & \cos(\theta) \end{bmatrix}\Bigg|_{\theta = \pi/(2T_r)}.$$

At the end of the $T = T_p + T_r$ time steps, the process lies along the *readout dimension* (also referred to as the output-potent direction in the literature), Fig. 4A (trajectories are shown as solid curves). The projections of $\mathbf{x}(t)$ onto the preparatory axis and read axis are shown in Fig. 4B.

The second system is a scalar process $y = \{y(t)\}$ that represents the downstream motor readout of first system and is a noisy projection of $\mathbf{x}(t)$ onto the readout axis:

$$\text{Readout projection} \qquad y(t) = \boldsymbol{e}_2^\top \mathbf{x}(t) + v(t), \qquad (12)$$

where $v(t)$ is additive Gaussian noise. Similar to the scalar example from the previous section, the two process $\{\mathbf{x}(t)\}$ and $\{y(t)\}$ have an important distinction: if the values of $\mathbf{x}(t)$ for $t \in [0, T_p]$ are known, then we have much more information about the values of $\mathbf{x}(t)$ for $t \in [T_p, T_p + T_r]$. On the other hand, if the values of $y(t)$ for $t \in [0, T_p]$ are known, then we have no additional information about the values of $y(t)$ for $t \in [T_p, T_p + T_r]$.

We apply Procrustes shape distance, 2-SSD, DSA and 2-Causal OT distance to compare the process $\mathbf{x}$ and $y$ (since $y$ is a one-dimensional process, we embed it in $\mathbb{R}^2$ via the mapping $y \mapsto \boldsymbol{e}_2 y$). When preparatory activity is large, we find that Procrustes shape distance, 2-SSD, DSA and 2-Causal OT distance can each differentiate the two systems (Fig. 4D, vertical red line). However, often preparatory activity is small relative to motor activity (Churchland et al., 2012; Kaufman et al., 2016). We can model this by scaling the preparatory axis by a constant $\gamma > 0$: $(x_1, x_2) \mapsto (\gamma x_1, x_2)$. This results in smaller preparatory activity (Fig. 4A, trajectories are dashed curves), consistent with experimental observations. In this case, when preparatory activity is small, we see that Procrustes shape distance and SSD are now much smaller whereas Causal OT and DSA distances do not significantly change (Fig. 4D, vertical dashed blue line versus vertical red solid line). Moreover, as preparatory activity shrinks $\gamma \to 0$ (output potent/null ratio grows), then Procrustes shape distance and SSD converge to zero, while Causal OT and DSA distances plateau (Fig. 4D). Importantly, this implies that Procrustes shape distance and SSD cannot capture differences between a system with small preparatory activity and motor activity and a system with only motor activity, while Causal OT and DSA distances can distinguish these two systems. In this way, both Causal OT and DSA distances capture important differences in the across-time statistical structure of the two processes that is not captured by Procrustes shape distance and SSD.

### 4.3 LATENT DIFFUSION MODELS FOR IMAGE GENERATION

Denoising diffusion models (Sohl-Dickstein et al., 2015; Song & Ermon, 2019; Ho et al., 2020; Kadkhodaie & Simoncelli, 2021) are powerful conditional and unconditional generative models

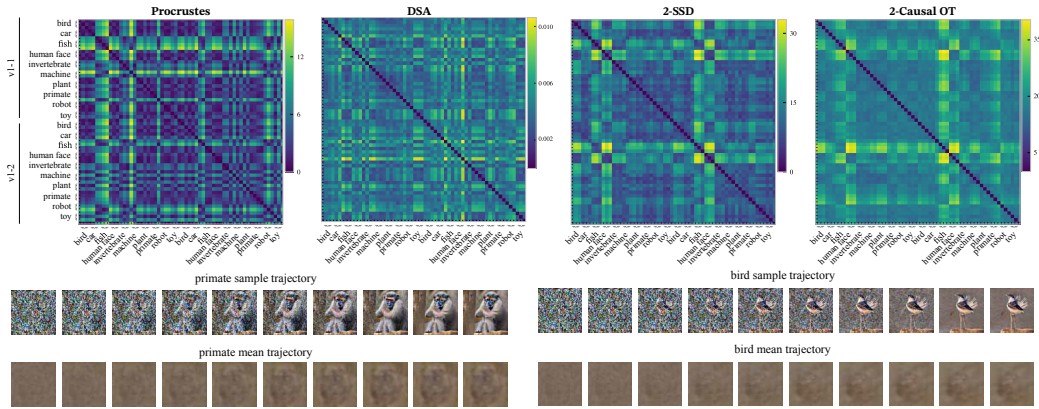

Figure 5: Comparisons of sample trajectories generated from text-to-image stable diffusion models `v1-1`, `v1-2` for 10 different prompts. **Top row:** Distance matrices show the estimated pairwise distances between the conditional processes. **Bottom row:** We show one example of a random trajectory from the "primate" and "bird" prompts and their corresponding mean trajectories decoded into the pixel space. While individual trajectories for each prompt category generate an image with features corresponding to that category, the mean trajectories decoded into the pixel space do not contain prompt-related visual features.

with a wide range of applications (Yang et al., 2023). While there are well-established methods for probing internal representations of deterministic and static neural networks, these methods have not (to our knowledge) been widely applied to diffusion models (Klabunde et al., 2023). Furthermore, the forward and reverse diffusion processes leveraged by these generative models are stochastic dynamical systems, making them ideal candidates for the methodology we have developed.

Here, we investigate the extent to which the metrics (Procrustes, DSA, SSD, Causal OT) can distinguish stochastic dynamical trajectories generated by a diffusion model. Given the nonlinearity of diffusion processes, it's certainly possible that none of these measures can capture meaningful similarities and differences between these trajectories. It's also possible that these methods can capture meaningful similarities and differences given enough samples, but the high-dimensionality of these systems means that estimation of the statistics would require an impractical number of sample trajectories.

To test our framework, we consider two pretrained text-to-image latent diffusion models (`v1-1` and `v1-2`)[3] trained to generate text-conditional images from noise (Rombach et al., 2022). Given a text prompt, each model generates a $2^{14}$-dimensional stochastic latent trajectory $\mathbf{x}^{\text{model, prompt}}$, whose terminal point is decoded into an image (with dimensions $256 \times 256 \times 3$). In particular, each model and prompt corresponds to a distinct stochastic dynamical system. Therefore, we expect that for each model, $d(\mathbf{x}^{\text{model, prompt}}, \mathbf{x}^{\text{model, prompt}}) = 0$ for the same prompts, and potentially $d(\mathbf{x}^{\text{model, prompt A}}, \mathbf{x}^{\text{model, prompt B}}) > 0$ for different prompts (though this is less clear due to the nuisance transform). Moreover, the training dataset used for `v1-1` was a subset of the dataset used for `v1-2`, so we might expect their dynamics to be similar, in which case we should find that $d(\mathbf{x}^{\text{v1-1, prompt}}, \mathbf{x}^{\text{v1-2, prompt}}) \approx 0$.

We prompted each diffusion model with 10 different prompts from both animate and inanimate sources: "bird", "car", "fish", "human face", "invertebrate", "machine", "plant", "primate", "robot", "toy". These prompts were inspired by the stimulus categories used to probe image representations in humans (fMRI) and non-human primates (single-cell recordings) (Kriegeskorte et al., 2008). For each prompt and each model, we generated 60 latent trajectories and decoded those trajectories into the image space (examples of two decoded trajectories are shown in Fig. 5 and for several other trajectories in Fig. 7 of the supplement). We repeated this process for 3 random seeds to use the within-category distances as a baseline. This provided 60 datasets (2 diffusion models, 10 prompts, 3 seeds per prompt) each containing 60 latent trajectories.

---

[3]Models were taken from https://huggingface.co/CompVis.

For each pair of processes, we estimated the Procrustes shape distance, DSA distance, SSD and Causal OT distance (Fig. 5). Estimating the distances in the $2^{14}$-dimensional latent space is intractable, so we first projected each set of latent trajectories onto its top 10 principal components (PCs) before computing the distances between the 10-dimensional stochastic trajectories. To ensure that 10 PCs are sufficient for recapitulating the data, we recomputed the pairwise distances as the number of PCs vary from 2 to 20 and showed that the distances indeed converge (Supp. Fig. 6).

Visual inspection of the pairwise distance matrices reveals that SSD and Causal OT capture structured relations between the trajectories that are not captured by Procrustes distance or DSA. First, both the SSD and Causal OT distance matrices have a $3 \times 3$ block structure, while the Procrustes and DSA distance matrices do not. The $3 \times 3$ block structure reflects the expected smaller within-category distances than between-category distances; that is, $d(\mathbf{x}^A, \mathbf{x}^A) < d(\mathbf{x}^A, \mathbf{x}^B)$ for $A \neq B$. This structure also suggests that the number of trials used for computing distances provides sufficient statistical power for capturing meaningful distances. Second, SSD and Causal OT appear to capture similarities in the representations between models v1-1 and v1-2, where as Procrustes does not. Specifically, SSD and Causal OT distance appear to mainly depend on the prompt (i.e., they are invariant to the choice of model). This is reflected in the fact that the four quadrants of the distance matrices have similar apparent structure. On the other hand, this structure is not apparent in the Procrustes or DSA distance matrices. Together, these results suggest that it is important to account for correlation structure when comparing the similarities of these latent stochastic trajectories.

To better understand why Procrustes distance fails to capture structured differences between processes, we investigated whether the mean latent trajectories contain any visual information about the prompt categories. To this end, for each model and category, we passed the mean latent trajectory through the decoder and visually inspected the resulting image (bottom row of Fig. 5, also Supp. Fig. 7). The generated images are largely indistinguishable from each other suggesting that Procrustes distance fails to capture important stochastic aspects of the internal representations.

There are some other interesting aspects related to the distance matrices. First, there is an apparent hierarchy of representational distances encoded in the block diagonals which opens up interesting questions about the differences between generative and discriminative models of vision for the cognitive neuroscience community (e.g., see Supp. Fig. 8). Causal OT distances are larger and appear less variable in the block diagonals suggesting that the full covariance structure is informative for computing shape distances. Importantly, distances that capture the dynamic and stochastic aspects of representations may be useful tools for investigating diffusion models.

## 5 DISCUSSION

In this work, we introduced Causal OT distance for comparing noisy neural dynamics. The distance is a natural extension of existing shape metrics (Procrustes and SSD) that accounts for both the stochastic and dynamic aspects of neural responses, while also respecting temporal causality. We applied our method to compare simple models of neural responses in motor systems and found that Causal OT distance can distinguish between neural responses encoding preparatory dynamics and motor outputs versus neural responses that only encode motor outputs. We also applied our method to compare representations in latent diffusion models whose generation process is a stochastic dynamical system. We found that Causal OT distances can differentiate generative processes while Procrustes distance and DSA cannot (though SSD can), emphasizing the need to consider stochasticity when analyzing such systems.

There are important limitations. First, the Causal OT distance only captures first- and second-order moments, which is sufficient for distinguishing Gaussian processes; however, it is not guaranteed to capture differences between non-Gaussian processes. Second, estimating first and second moments can require a significant number of sample trajectories. For example, an $N$-dimensional trajectory with $T$ time points has an $NT \times NT$ covariance matrix. When $N$ or $T$ are large, accurate estimation of this covariance matrix can require more sample trajectories than are available. Theoretical analysis of estimating shape distances and other representational dissimilarity measures from finite data remains an active area of study (Pospisil et al., 2023). One promising approach is to impose priors on the stochastic processes to reduce the number of trials required to estimate the first- and second-order moments (Duncker et al., 2019; Nejatbakhsh et al., 2023; Geadah et al., 2025).

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

# A    CAUSAL OPTIMAL TRANSPORT DISTANCE BETWEEN GAUSSIAN PROCESSES

In this section, we provide a first-principles derivation of the Causal OT distance between two processes defined in equation 8. The distance can be derived as an optimal transport distance between two Gaussian processes $\mathbf{x}$ and $\mathbf{y}$ where the transport cost is an $L^2$-distance and the admissible couplings correspond the set of so-called *causal synchronous couplings*. These are *synchronous* couplings because both Gaussian processes are driven by a common white noise process $\mathbf{w} = \{\mathbf{w}(t)\}$, and they are *causal* in the sense that at any given time point, the current values of the Gaussian processes $\mathbf{x}(t)$ and $\mathbf{y}(t)$ do not depend on the future values of the white noise process $\mathbf{w}(t+1), \mathbf{w}(t+2), \ldots$.

This distance is closely related to bi-causal optimal transport distances (also referred to as adapted Wasserstein distances) between Gaussian processes (Lassalle, 2018; Backhoff et al., 2017; Gunasingam & Wong, 2024). These are optimal transport distances in which the couplings between the processes are required to satisfy a "bi-causal" property, which can be interpreted as "the past of process 1 is independent of the future of process 2, conditioned on the past of process 2" and, conversely, "the past of process 2 is independent of the future of process 1, conditioned on the past of process 1". Here our couplings are defined in terms of causal synchronous couplings rather than bi-causal couplings. These two notions of coupling are equivalent in the scalar setting (Backhoff-Veraguas et al., 2022); however, we are unaware of a proof of this in the multi-dimensional setting.

Suppose $\mathbf{x}$ and $\mathbf{y}$ are $N$-dimensional Gaussian processes defined for $t = 1, \ldots, T$. Let $\mathbf{w} = \{\mathbf{w}(t)\}$ be a sequence of i.i.d. $N$-dimensional standard normal vectors (i.e., a white noise process). We define a *causal synchronous coupling* between $\mathbf{x}$ and $\mathbf{y}$ as a pair of processes $(\hat{\mathbf{x}}, \hat{\mathbf{y}})$ that satisfy the following criteria:

- **Synchronous coupling condition:**
    - (i) there is a function that maps the white noise process $\mathbf{w}$ to $(\hat{\mathbf{x}}, \hat{\mathbf{y}})$;
    - (ii) $\hat{\mathbf{x}}$ is equal in distribution to $\mathbf{x}$ and $\hat{\mathbf{y}}$ is equal in distribution to $\mathbf{y}$.
- **Causality condition:**
    - (iii) for each $t < T$, there is a function that maps $(\mathbf{w}(1), \ldots, \mathbf{w}(t))$ to $(\hat{\mathbf{x}}(t), \hat{\mathbf{y}}(t))$.

We now construct a family of causal synchronous couplings. Let $\vec{\mathbf{x}}$ be the $NT$-dimensional Gaussian random vector obtained by concatenating $\mathbf{x}(1), \ldots, \mathbf{x}(T)$:

$$\vec{\mathbf{x}} := \begin{bmatrix} \mathbf{x}(1) \\ \vdots \\ \mathbf{x}(T) \end{bmatrix} \sim \mathcal{N}\left(\vec{\boldsymbol{m}}_x, \boldsymbol{C}_x\right),$$

where $\vec{\boldsymbol{m}}_x$ and $\boldsymbol{C}_x$ denote the mean and covariance of the concatenated vector $\vec{\mathbf{x}}$. Let $\boldsymbol{C}_x = \boldsymbol{L}_x \boldsymbol{L}_x^\top$ be the Cholesky decomposition of the covariance matrix $\boldsymbol{C}_x$. If $\boldsymbol{C}_x$ is full rank, then the decomposition is unique. If $\boldsymbol{C}_x$ is rank $R < N$, then we take $\boldsymbol{L}_x$ to be the unique lower-triangular matrix with exactly $R$ positive diagonal elements and $N - R$ columns containing all zeros (Gentle, 2012). Given an $N$-dimensional *driving white noise process* $\mathbf{w} = \{\mathbf{w}(1), \ldots, \mathbf{w}(T)\}$—i.e., $\mathbf{w}(t)$ are i.i.d. $N$-dimensional Gaussian random vectors with identity covariance—define the process $\hat{\mathbf{x}} = \{\hat{\mathbf{x}}(t)\}$ by

$$\begin{bmatrix} \hat{\mathbf{x}}(1) \\ \vdots \\ \hat{\mathbf{x}}(T) \end{bmatrix} = f^x(\mathbf{w}) := \vec{\boldsymbol{m}}_x + \boldsymbol{L}_x \vec{\mathbf{w}}, \qquad\qquad \vec{\mathbf{w}} := \begin{bmatrix} \mathbf{w}(1) \\ \vdots \\ \mathbf{w}(T) \end{bmatrix}.$$

Since the distribution of a Gaussian process can be completely characterized in terms of its first two moments, the process $\hat{\mathbf{x}}$ is equal in *distribution* to the process $\mathbf{x}$. Furthermore, by construction $\hat{\mathbf{x}}$ is a continuous function of the driving white noise process $\mathbf{w}$ and, since $\boldsymbol{L}_x$ is lower triangular, for any $t = 1, \ldots, T-1$,

$$\begin{bmatrix} \hat{\mathbf{x}}(1) \\ \vdots \\ \hat{\mathbf{x}}(t) \end{bmatrix} = f_t^x(\mathbf{w}_t) := \vec{\boldsymbol{m}}_{x,t} + \boldsymbol{L}_{x,t} \vec{\mathbf{w}}_t, \qquad\qquad \vec{\mathbf{w}}_t := \begin{bmatrix} \mathbf{w}(1) \\ \vdots \\ \mathbf{w}(t) \end{bmatrix}, \qquad (13)$$

where $\vec{m}_{x,t}$ denotes $Nt$-dimensional vector that is equal to the first $Nt$ coordinates of $\vec{m}_x$, and $\boldsymbol{L}_{x,t}$ denotes the $Nt \times Nt$ matrix that is equal to the first $Nt$ rows and $Nt$ columns of $\boldsymbol{L}_x$. We say that $\hat{\mathbf{x}}$ is *adapted* to the driving white noise process $\mathbf{w}$ since equation 13 holds; that is, for each $t = 1, \ldots, T-1$, the vector $\hat{\mathbf{x}}(t)$ is a function of the first $t$ steps of the white noise process $\mathbf{w}$. As a corollary, $\hat{\mathbf{x}}(t)$ is independent of the future values of the white noise process $\mathbf{w}(t+1), \ldots, \mathbf{w}(T)$.

Let $f^y(\cdot)$ and $f^y_t(\cdot)$ be defined as above but with $x$'s replaced by $y$'s. Then $\hat{\mathbf{x}} = f^x(\mathbf{w})$ and $\hat{\mathbf{y}} = f^y(\mathbf{w})$ is a causal synchronous coupling between $\mathbf{x}$ and $\mathbf{y}$. In general, we consider causal synchronous couplings of the form:

$$\hat{\mathbf{x}} = f^x(\mathbf{w}), \qquad\qquad \hat{\mathbf{y}} = f^y(\boldsymbol{R}_1\mathbf{w}(1), \ldots, \boldsymbol{R}_T\mathbf{w}(T)),$$

where $\boldsymbol{R}_1, \ldots, \boldsymbol{R}_T$ are $N \times N$ orthogonal matrices. Since the distribution of white noise is invariant these transformations, these define synchronous couplings of $\mathbf{x}$ and $\mathbf{y}$. Furthermore, the rotations do not rotate the white noise process in *time*, so they define causal synchronous couplings of $\mathbf{x}$ and $\mathbf{y}$.

Having defined our admissible couplings, we now define the *Causal OT* distance as

$$d_{\text{causal}}(\mathbf{x}, \mathbf{y}) := \min_{\boldsymbol{Q}, \boldsymbol{R}_1, \ldots, \boldsymbol{R}_T \in O(N)} \mathbb{E}\left[\left\| f^x(\mathbf{w}) - (\boldsymbol{I}_T \otimes \boldsymbol{Q}) f^y(\boldsymbol{R}_1\mathbf{w}(1), \ldots, \boldsymbol{R}_T\mathbf{w}(T)) \right\|^2\right], \quad (14)$$

where the minimization is over nuisance transformations $\boldsymbol{Q} \in O(N)$ and over spatial rotations $\boldsymbol{R}_1, \ldots, \boldsymbol{R}_T \in O(N)$ of the noise. Substituting in with the formulas for $f^x$ and $f^y$, we see that

$$\mathbb{E}\left[\left\| f^x(\mathbf{w}) - (\boldsymbol{I}_T \otimes \boldsymbol{Q}) f^y(\boldsymbol{R}_1\mathbf{w}(1), \ldots, \boldsymbol{R}_T\mathbf{w}(T)) \right\|^2\right]$$
$$= \left\| \vec{m}_x - (\boldsymbol{I}_T \otimes \boldsymbol{Q})\vec{m}_y \right\|^2 + \mathbb{E}\left[\left\| \boldsymbol{L}_x\vec{\mathbf{w}} - (\boldsymbol{I}_T \otimes \boldsymbol{Q})\boldsymbol{L}_y\text{diag}(\boldsymbol{R}_1, \ldots, \boldsymbol{R}_T)\vec{\mathbf{w}} \right\|^2\right]$$
$$= \left\| \vec{m}_x - (\boldsymbol{I}_T \otimes \boldsymbol{Q})\vec{m}_y \right\|^2 + \left\| \boldsymbol{L}_x - (\boldsymbol{I}_T \otimes \boldsymbol{Q})\boldsymbol{L}_y\text{diag}(\boldsymbol{R}_1, \ldots, \boldsymbol{R}_T) \right\|_F^2.$$

Substituting this expression back into equation 14, we get

$$d_{\text{causal}}^2(\mathbf{x}, \mathbf{y}) = \min_{\boldsymbol{Q} \in O(N)} \left\{ \left\| \vec{m}_x - (\boldsymbol{I}_T \otimes \boldsymbol{Q})\vec{m}_y \right\|^2 + \mathcal{AB}_{N,T}^2(\boldsymbol{C}_x, (\boldsymbol{I}_T \otimes \boldsymbol{Q})\boldsymbol{C}_y(\boldsymbol{I}_T \otimes \boldsymbol{Q}^\top)) \right\},$$

where $\mathcal{AB}_{N,T}(\cdot, \cdot)$ is the *adapted Bures* distance between positive semidefinite matrices defined

$$\mathcal{AB}_{N,T}(\boldsymbol{A}, \boldsymbol{B}) := \min_{\boldsymbol{R}_1, \ldots, \boldsymbol{R}_T \in O(N)} \left\| \boldsymbol{L}_A - \boldsymbol{L}_B\text{diag}(\boldsymbol{R}_1, \ldots, \boldsymbol{R}_T) \right\|_F,$$

where $\boldsymbol{A} = \boldsymbol{L}_A\boldsymbol{L}_A^\top$ and $\boldsymbol{B} = \boldsymbol{L}_A\boldsymbol{L}_A^\top$ are the Cholesky decompositions. In the scalar setting, this reduces to the adapted Bures distance defined in (Gunasingam & Wong, 2024). While this metric is motivated as a distance between Gaussian processes, it defines a proper (pseudo-)metric between any stochastic processes in terms of their first- and second-order statistics.

It's worth noting that if we relax the *causality condition* then we can consider a larger set of (acausal) synchronous couplings:

$$\hat{\mathbf{x}} = f^x(\mathbf{w}), \qquad\qquad \hat{\mathbf{y}} = f^y(\Phi_{\boldsymbol{U}}(\mathbf{w})),$$

where $\boldsymbol{U} \in O(NT)$ is any $NT \times NT$ rotation and

$$\Phi_{\boldsymbol{U}}(\mathbf{w}) := \boldsymbol{U} \begin{bmatrix} \mathbf{w}(1) \\ \vdots \\ \mathbf{w}(T) \end{bmatrix}$$

is a rotation of the white noise process that leaves its distribution invariant. Minimizing the $L^2$-distance over all such couplings results in the following distance, which is equal to the Wasserstein distance (with $p = 2$) between Gaussian random vectors (Mallasto & Feragen, 2017) up to a nui-

sance transformation:

$$d^2_{\text{Wasserstein}}(\mathbf{x}, \mathbf{y}) = \min_{\boldsymbol{Q} \in O(N)} \min_{\boldsymbol{U} \in O(NT)} \mathbb{E} \left[ \| f^x(\mathbf{w}) - (\boldsymbol{I}_T \otimes \boldsymbol{Q}) f^y(\Phi_{\boldsymbol{U}}(\mathbf{w})) \|^2 \right]$$

$$= \min_{\boldsymbol{Q} \in O(N)} \left\{ \| \vec{\boldsymbol{m}}_x - (\boldsymbol{I}_T \otimes \boldsymbol{Q}) \vec{\boldsymbol{m}}_y \|^2 + \min_{\boldsymbol{U} \in O(NT)} \mathbb{E} \left[ \| \boldsymbol{L}_x \vec{\mathbf{w}} - (\boldsymbol{I}_T \otimes \boldsymbol{Q}) \boldsymbol{L}_y \boldsymbol{U} \vec{\mathbf{w}} \|^2 \right] \right\}$$

$$= \min_{\boldsymbol{Q} \in O(N)} \left\{ \| \vec{\boldsymbol{m}}_x - (\boldsymbol{I}_T \otimes \boldsymbol{Q}) \vec{\boldsymbol{m}}_y \|^2 + \min_{\boldsymbol{U} \in O(NT)} \| \boldsymbol{L}_x - (\boldsymbol{I}_T \otimes \boldsymbol{Q}) \boldsymbol{L}_y \boldsymbol{U} \|_F^2 \right\}$$

$$= \min_{\boldsymbol{Q} \in O(N)} \left\{ \| \vec{\boldsymbol{m}}_x - (\boldsymbol{I}_T \otimes \boldsymbol{Q}) \vec{\boldsymbol{m}}_y \|^2 + \min_{\boldsymbol{U} \in O(NT)} \left\| (\boldsymbol{C}_x)^{1/2} - (\boldsymbol{I}_T \otimes \boldsymbol{Q})(\boldsymbol{C}_y)^{1/2} \boldsymbol{U} \right\|_F^2 \right\}$$

$$= \min_{\boldsymbol{Q} \in O(N)} \left\{ \| \vec{\boldsymbol{m}}_x - (\boldsymbol{I}_T \otimes \boldsymbol{Q}) \vec{\boldsymbol{m}}_y \|^2 + \mathcal{B}^2(\boldsymbol{C}_x, (\boldsymbol{I}_T \otimes \boldsymbol{Q}) \boldsymbol{C}_y (\boldsymbol{I}_T \otimes \boldsymbol{Q}^\top)) \right\}$$

where $\mathcal{B}(\cdot, \cdot)$ is the Bures distance on positive semidefinite matrices:

$$\mathcal{B}(\boldsymbol{A}, \boldsymbol{B}) := \min_{\boldsymbol{U} \in O(NT)} \| \boldsymbol{A}^{1/2} - \boldsymbol{B}^{1/2} \boldsymbol{U} \|_F.$$

## B    ALTERNATING MINIMIZATION ALGORITHM

Suppose $\mathbf{x} = \{\mathbf{x}(t)\}_{t=1,\ldots,T}$ is an $N_x$-dimensional Gaussian process and $\mathbf{y} = \{\mathbf{y}(t)\}_{t=1,\ldots,T}$ is an $N_y$-dimensional Gaussian process. If $N_x \neq N_y$, we can pad the lower-dimensional process with zeros so that they are both $N$-dimensional processes, where $N := \max(N_x, N_y)$. For example, if $N_y < N_x$, then we can embed $\mathbf{y}(t)$ into $\mathbb{R}^N$ via the linear transformation

$$\mathbf{y}(t) \mapsto \begin{bmatrix} \mathbf{y}(t) \\ \mathbf{0} \end{bmatrix},$$

where $\mathbf{0}$ is a $(N - N_y)$-dimensional vector of zeros. For the remainder of this section, we assume that $N = N_x = N_y$.

From equation 8, we have that causal OT distance is equal to

$$\min_{\boldsymbol{Q}, \boldsymbol{R}_1, \ldots, \boldsymbol{R}_T} \left\{ \|\vec{\boldsymbol{m}}_x - (\boldsymbol{I}_T \otimes \boldsymbol{Q})\vec{\boldsymbol{m}}_y\|^2 + \|\boldsymbol{L}_x - (\boldsymbol{I}_T \otimes \boldsymbol{Q})\boldsymbol{L}_y \text{diag}(\boldsymbol{R}_1, \ldots, \boldsymbol{R}_T)\|_F^2 \right\}. \tag{15}$$

We solve this via an alternating minimization algorithm. For an $NT \times NT$ matrix $\boldsymbol{L}$ and $1 \leq s, t \leq T$, let $\boldsymbol{L}_{st}$ denote the $(s, t)^{\text{th}}$ $N \times N$ block of $\boldsymbol{L}$. We first rewrite the second term as:

$$\|\boldsymbol{L}_x - (\boldsymbol{I} \otimes \boldsymbol{Q})\boldsymbol{L}_y \text{diag}(\boldsymbol{R}_1, \ldots, \boldsymbol{R}_T)\|_F^2 = \sum_{s=1}^{T} \sum_{t=1}^{t} \|\boldsymbol{L}_{x,st} - \boldsymbol{Q}\boldsymbol{L}_{y,st}\boldsymbol{R}_t\|_F^2$$

$$= \sum_{t=1}^{T} \sum_{s=t}^{T} \|\boldsymbol{L}_{x,st} - \boldsymbol{Q}\boldsymbol{L}_{y,st}\boldsymbol{R}_t\|_F^2$$

$$= \sum_{t=1}^{T} \left\| \begin{bmatrix} \boldsymbol{L}_{x,tt} \\ \vdots \\ \boldsymbol{L}_{x,Tt} \end{bmatrix} - \begin{bmatrix} \boldsymbol{Q}\boldsymbol{L}_{y,tt} \\ \vdots \\ \boldsymbol{Q}\boldsymbol{L}_{y,Tt} \end{bmatrix} \boldsymbol{R}_t \right\|_F^2.$$

Then optimizing over $\boldsymbol{R}_t$ yields:

$$\boldsymbol{R}_t = \arg\min_{\boldsymbol{R}} \left\| \begin{bmatrix} \boldsymbol{L}_{x,tt} \\ \vdots \\ \boldsymbol{L}_{x,Tt} \end{bmatrix} - \begin{bmatrix} \boldsymbol{Q}\boldsymbol{L}_{y,tt} \\ \vdots \\ \boldsymbol{Q}\boldsymbol{L}_{y,Tt} \end{bmatrix} \boldsymbol{R} \right\|_F^2$$

Alternatively, we can express the sum in equation 15 as

$$\|\vec{\boldsymbol{m}}_x - (\boldsymbol{I} \otimes \boldsymbol{Q})\vec{\boldsymbol{m}}_y\|^2 + \|\boldsymbol{L}_x - (\boldsymbol{I} \otimes \boldsymbol{Q})\boldsymbol{L}_y \text{diag}(\boldsymbol{R}_1, \ldots, \boldsymbol{R}_T)\|_F^2$$

$$= \left\| \begin{bmatrix} \boldsymbol{m}_{x,1}^\top \\ \vdots \\ \boldsymbol{m}_{x,T}^\top \end{bmatrix} - \begin{bmatrix} \boldsymbol{m}_{y,1}^\top \\ \vdots \\ \boldsymbol{m}_{y,T}^\top \end{bmatrix} \boldsymbol{Q}^\top \right\|_F^2 + \left\| \begin{bmatrix} \boldsymbol{L}_{x,11}^\top \\ \vdots \\ \boldsymbol{L}_{x,TT}^\top \end{bmatrix} - \begin{bmatrix} \boldsymbol{R}_1^\top \boldsymbol{L}_{y,11}^\top \\ \vdots \\ \boldsymbol{R}_T^\top \boldsymbol{L}_{y,TT}^\top \end{bmatrix} \boldsymbol{Q}^\top \right\|_F^2$$

$$= \left\| \begin{bmatrix} \boldsymbol{m}_{x,1}^\top \\ \vdots \\ \boldsymbol{m}_{x,T}^\top \\ \boldsymbol{L}_{x,11}^\top \\ \vdots \\ \boldsymbol{L}_{x,TT}^\top \end{bmatrix} - \begin{bmatrix} \boldsymbol{m}_{y,1}^\top \\ \vdots \\ \boldsymbol{m}_{y,T}^\top \\ \boldsymbol{R}_1^\top \boldsymbol{L}_{y,11}^\top \\ \vdots \\ \boldsymbol{R}_T^\top \boldsymbol{L}_{y,TT}^\top \end{bmatrix} \boldsymbol{Q}^\top \right\|_F^2.$$

Then optimizing over $\boldsymbol{Q}$ yields

$$\boldsymbol{Q} = \arg\min_{\boldsymbol{Q}} \left\| \begin{bmatrix} \boldsymbol{m}_{x,1}^\top \\ \vdots \\ \boldsymbol{m}_{x,T}^\top \\ \boldsymbol{L}_{x,11}^\top \\ \vdots \\ \boldsymbol{L}_{x,TT}^\top \end{bmatrix} - \begin{bmatrix} \boldsymbol{m}_{y,1}^\top \\ \vdots \\ \boldsymbol{m}_{y,T}^\top \\ \boldsymbol{R}_1^\top \boldsymbol{L}_{y,11}^\top \\ \vdots \\ \boldsymbol{R}_T^\top \boldsymbol{L}_{y,TT}^\top \end{bmatrix} \boldsymbol{Q}^\top \right\|_F^2.$$

We summarize the previous steps into an algorithm. Given

$$\vec{m} = \begin{bmatrix} \boldsymbol{m}_1 \\ \vdots \\ \boldsymbol{m}_T \end{bmatrix}, \qquad \boldsymbol{L} = \begin{bmatrix} \boldsymbol{L}_{11} & & \\ \vdots & \ddots & \\ \boldsymbol{L}_{T1} & \cdots & \boldsymbol{L}_{TT} \end{bmatrix}$$

define the functions

$$F_t(\boldsymbol{L}) := \begin{bmatrix} \boldsymbol{L}_{tt} \\ \vdots \\ \boldsymbol{L}_{Tt} \end{bmatrix}, \qquad G(\vec{m}, \boldsymbol{L}) := \begin{bmatrix} \boldsymbol{m}_1^\top \\ \vdots \\ \boldsymbol{m}_T^\top \\ \boldsymbol{L}_{11}^\top \\ \boldsymbol{L}_{21}^\top \\ \boldsymbol{L}_{22}^\top \\ \vdots \\ \boldsymbol{L}_{TT}^\top \end{bmatrix}.$$

Let Cholesky$(\cdot)$ be a function that maps a positive semidefinite matrix to its Cholesky decomposition. When the matrix is full rank, this decomposition is unique. When the matrix is rank $R < N$, we select the unique decomposition with $R$ positive diagonal elements and $N - R$ columns whose entries are all zero.

---

**Algorithm 1:** Alternating minimization for computing causal OT distance

---

1: **input:** $0 \le \alpha \le 2$, $\vec{m}_x \in \mathbb{R}^{NT}$, $\vec{m}_y \in \mathbb{R}^{NT}$, $\boldsymbol{C}_x \in \mathcal{S}_{++}^{NT}$ and $\boldsymbol{C}_y \in \mathcal{S}_{++}^{NT}$.
2: **initialize:** $\boldsymbol{Q} \in O(N)$
3: $\boldsymbol{L}_x \leftarrow \text{Cholesky}(\boldsymbol{C}_x)$
4: $\boldsymbol{L}_y \leftarrow \text{Cholesky}(\boldsymbol{C}_y)$
5: **while** not converged **do**
6:     **for** $t = 1, \ldots, T$ **do**
7:         $\boldsymbol{M}_{x,t} \leftarrow F_t(\boldsymbol{L}_x)$
8:         $\boldsymbol{M}_{y,t} \leftarrow F_t((\boldsymbol{I}_y \otimes \boldsymbol{R})\boldsymbol{L}_y)$
9:         $\boldsymbol{R}_t \leftarrow \arg\min_{\boldsymbol{R} \in O(N)} \|\boldsymbol{M}_{x,t} - \boldsymbol{M}_{y,t}\boldsymbol{R}\|_F^2$
10:     **end for**
11:     $\boldsymbol{N}_x \leftarrow G(\vec{m}_x, \boldsymbol{L}_x)$
12:     $\boldsymbol{N}_y \leftarrow g(\vec{m}_y, \boldsymbol{L}_y\text{diag}(\boldsymbol{R}_1, \ldots, \boldsymbol{R}_T))$
13:     $\boldsymbol{Q}_y \leftarrow \arg\min_{\boldsymbol{Q} \in O(N)} \|\boldsymbol{N}_x - \boldsymbol{N}_y\boldsymbol{Q}^\top\|_F^2$
14: **end while**
15: $\text{dist} \leftarrow \sqrt{(2 - \alpha)\|\vec{m}_x - (\boldsymbol{I}_T \otimes \boldsymbol{Q})\vec{m}_y\|^2 + \alpha \left\|\boldsymbol{L}_x - (\boldsymbol{I}_T \otimes \boldsymbol{Q})\boldsymbol{L}_y\text{diag}(\boldsymbol{R}_1, \ldots, \boldsymbol{R}_T)\right\|_F^2}$
16: **return** dist

---

## C  SCALAR EXAMPLE

We derive the distances reported in Sec. 4.1. The means of $x$ and $y$ are identically zero so the Procrustes distance is zero. The covariance matrices of $(x(1), x(2))$ and $(y(1), y(2))$ satisfy

$$\boldsymbol{C}_x = \begin{bmatrix} \epsilon^2 & \epsilon\sigma \\ \epsilon\sigma & \sigma^2 \end{bmatrix} = \begin{bmatrix} \epsilon & 0 \\ \sigma & \sigma \end{bmatrix} \begin{bmatrix} \epsilon & \sigma \\ 0 & \sigma \end{bmatrix} = \boldsymbol{L}_x \boldsymbol{L}_x^\top,$$

$$\boldsymbol{C}_y = \begin{bmatrix} 0 & 0 \\ 0 & \sigma^2 \end{bmatrix} = \begin{bmatrix} 0 & 0 \\ 0 & \sigma \end{bmatrix} \begin{bmatrix} 0 & 0 \\ 0 & \sigma \end{bmatrix} = \boldsymbol{L}_y \boldsymbol{L}_y^\top.$$

From these expressions, we see that $d_{\text{SSD-1}}(x, y) = \epsilon$. Using the explicit formula for the Bures distance (Bhatia et al., 2019), the Wasserstein distance is

$$
\begin{aligned}
d_{\text{Wasserstein}}(x, y) &= \mathcal{B}(\boldsymbol{C}_x, \boldsymbol{C}_y) \\
&= \sqrt{\text{Tr}(\boldsymbol{C}_x) + \text{Tr}(\boldsymbol{C}_y) - 2\,\text{Tr}\left(\sqrt{\boldsymbol{C}_y}\boldsymbol{C}_x\sqrt{\boldsymbol{C}_y}\right)^{1/2}} \\
&= \sqrt{(\epsilon^2 + \sigma^2) + \sigma^2 - 2\sigma^2} \\
&= \epsilon,
\end{aligned}
$$

where we have used the fact that $\sqrt{\boldsymbol{C}_y} = \boldsymbol{L}_y$ in the third equality. Finally, by equation 8, the Causal OT-1 distance is

$$
\begin{aligned}
d_{\text{causal-1}}(x, y) &= \mathcal{AB}(\boldsymbol{C}_x, \boldsymbol{C}_y) \\
&= \min_{\alpha,\beta\in\{\pm1\}} \|\boldsymbol{L}_x - \boldsymbol{L}_y\text{diag}(\alpha, \beta)\|_F \\
&= \min_{\beta\in\{\pm1\}} \left\| \begin{bmatrix} \epsilon & 0 \\ \sigma & \sigma - \beta\sigma \end{bmatrix} \right\| \\
&= \sqrt{\epsilon^2 + \sigma^2}.
\end{aligned}
$$

# D  ADVERSARIAL TUNING OF NOISE CORRELATIONS AND INPUT

In this section we describe how to build adversarial examples of systems with distinct recurrent dynamics that share their marginal statistics. To achieve this, we will tune the input and noise correlations to these systems. Denoting a reference system that has marginal means $\boldsymbol{m}_t$ and marginal covariances $\boldsymbol{P}_t$ our goal is to build linear systems for arbitrary recurrent dynamics matrices $\boldsymbol{A}$ such that their marginal statistics are equivalent to the reference system.

To begin, we rewrite the dynamics of a linear system driven by inputs $\boldsymbol{b}_t$ and noise correlations $\boldsymbol{\Sigma}\boldsymbol{\Sigma}^\top$:

$$\boldsymbol{x}_{t+1} = \boldsymbol{x}_t + dt\boldsymbol{A}\boldsymbol{x}_t + dt\boldsymbol{b}_t + \sqrt{dt}\boldsymbol{\Sigma}_t\boldsymbol{\epsilon}_t$$
$$\mathbb{E}[\boldsymbol{x}_{t+1}] = (\boldsymbol{I} + dt\boldsymbol{A})\mathbb{E}[\boldsymbol{x}_t] + dt\boldsymbol{b}_t$$
$$\mathrm{Cov}(\boldsymbol{x}_{t+1}) = (\boldsymbol{I} + dt\boldsymbol{A})\mathrm{Cov}(\boldsymbol{x}_t)(\boldsymbol{I} + dt\boldsymbol{A}^\top) + dt\boldsymbol{\Sigma}_t\boldsymbol{\Sigma}_t^\top.$$

Therefore in order to have $\mathbb{E}[\boldsymbol{x}_t] = \boldsymbol{m}_t$ and $\mathrm{Cov}(\boldsymbol{x}_t) = \boldsymbol{P}_t$ we need to set $\boldsymbol{b}_t, \boldsymbol{\Sigma}_t$ according to the following equations.

$$\boldsymbol{b}_t = \frac{\boldsymbol{m}_{t+1} - (\boldsymbol{I} + dt\boldsymbol{A})\boldsymbol{m}_t}{dt}, \qquad\qquad \boldsymbol{b}_0 = \boldsymbol{m}_0$$
$$\boldsymbol{\Sigma}_t = \frac{\sqrt{\boldsymbol{P}_{t+1} - (\boldsymbol{I} + dt\boldsymbol{A})\boldsymbol{P}_t(\boldsymbol{I} + dt\boldsymbol{A}^\top)}}{\sqrt{dt}}, \qquad\qquad \boldsymbol{\Sigma}_0 = \sqrt{\boldsymbol{P}_0}$$

In Fig. 2 we first generated data from the Saddle dynamics since it is the only nonlinear model among the 3. We then used its marginal means and covariances to tune the inputs and noise correlations for other two systems (Line and Point attractor).

For completeness, we include the dynamical equations for the 3 systems below:

- **Saddle** (assuming that $\theta \in \{-0.1, 0.1\}$ is the coherency level):

$$f_1(\boldsymbol{x}) = -0.6\boldsymbol{x}_1^3 + 2\boldsymbol{x}_1 + \theta, \quad f_2(\boldsymbol{x}) = -\boldsymbol{x}_2 + \theta,$$
$$\frac{d\boldsymbol{x}}{dt} = f(\boldsymbol{x}) + \boldsymbol{\epsilon}_t$$

- **Point Attractor** (assuming the linear dynamics mentioned above with $\boldsymbol{A}$ dynamics matrix):

$$\boldsymbol{A} = \begin{bmatrix} -0.5 & 0 \\ 0 & -1 \end{bmatrix}$$

- **Line Attractor** (assuming the linear dynamics mentioned above with $\boldsymbol{A}$ dynamics matrix and $\mathrm{Rot}(\phi)$ is the rotation matrix for angle $\phi$):

$$\boldsymbol{A} = \begin{bmatrix} 0 & 0 \\ 0.7 & -1 \end{bmatrix} \mathrm{Rot}(\phi), \quad \phi = 45°$$

# E DETAILS OF THE DIFFUSION MODEL EXPERIMENT

Stable diffusion for text-to-image generation are latent diffusion models that consist of a few components. We denote images by random variable $\mathbf{s}$, text by random variable $\mathbf{p}$, and latent diffusion process by random variable $\mathbf{x}_{1:T}$. An encoder $\mathcal{E}$ maps images into the latent space and a decoder $\mathcal{D}$ maps the latent diffusion back into the image space. Text prompts are mapped into a fixed-length embedding $\tau_\theta(\mathbf{p})$ that can be used for conditioning the latent diffusion model. The text embedding uses an architecture that is well-suited for the text modality.

Separately, a time-conditioned UNet architecture is used to run the backward diffusion process for denoising images in the latent space. We denote the denoiser by $\epsilon_\theta(\mathbf{x}_t, t)$. Conditioning on text is performed via mapping the text representation to intermediate layers of the denoising UNet using a cross-attention mechanism (see Rombach et al. (2022) for details). Finally, the conditional diffusion process cost function $\mathcal{L}$ is used for the joint optimization of the denoiser and text embedding:

$$\mathcal{L} = \mathbb{E}_{\mathcal{E}, \mathbf{p}, \epsilon \sim \mathcal{N}(\mathbf{0}, \boldsymbol{I}), t} \left[ \left\| \epsilon - \epsilon_\theta(\mathbf{x}_t, t, \tau_\theta(\mathbf{p})) \right\|_2^2 \right]$$

Once the components of the model are trained, the function $\epsilon_\theta$ provides an approximation of the score function $\nabla \log p(\mathbf{x}|\mathbf{p})$ for large $t$.

During the image generation, first the encoding of the text prompt is computed. Then starting from random iid noise in the latent space, the following model is run forward:

$$\mathbf{x}_{t+1} = \mathbf{x}_t + h_t(\epsilon_\theta(\mathbf{x}_t, t, \tau_\theta(\mathbf{p})) - \mathbf{x}_t) + \gamma_t \mathbf{z}_t.$$

In our experiments, we generated samples from this stochastic process conditioned on different prompts and computed pairwise distances between different conditional processes using Procrustes, DSA, SSD, and Causal OT distances. For DSA we chose the hyperparameters `n_delays = 9`, `rank = 10`. We fixed all the other hyperparameters to the following for this and all other DSA experiments: `delay_interval = 1`, `lr = 0.01`, `iters = 1000`. By computing these pairwise distances we showed that both SSD and Causal OT capture our desired properties. Here, we present more results validating our findings.

The size of the images are $256 \times 256 \times 3$ (3 RGB channels) and the size of the latent space is $64 \times 64 \times 4$. Therefore the latent representation provides a computational gain with a factor of approximately 16. Since the dimension of the latent space is still very large, we performed PCA on the latent trajectories to map it onto the top $k$ PCs. In Fig. 6 we show the dependence of the normalized distances on $k \in \{2, 4, 6, 8, 10, 15, 20\}$. We normalized the distances to the median value of each distance matrix to focus on the relative changes. The normalized distance changes are very small after $k = 6$ confirming that the distances are not an artifact of the lower-dimensional projection.

In addition, we include the decoded sample trajectories and mean trajectories from all prompt categories for both models `v1-1` and `v1-2` in Fig. 7. The decoded mean trajectories show very subtle signatures of the image category confirming that distances based on mean trajectories (e.g. Procrustes) are not sufficient to capture differences in the stochastic processes.

Finally, we included the MDS projection of all three pairwise distances in Fig. 8. This figure shows that in both SSD and Causal OT the conditional stochastic processes corresponding to the same prompt category for different seeds and diffusion models cluster together. This suggests that MDS embedding of the stochastic process distances is meaningful and potentially relevant for the cognitive neuroscience community.

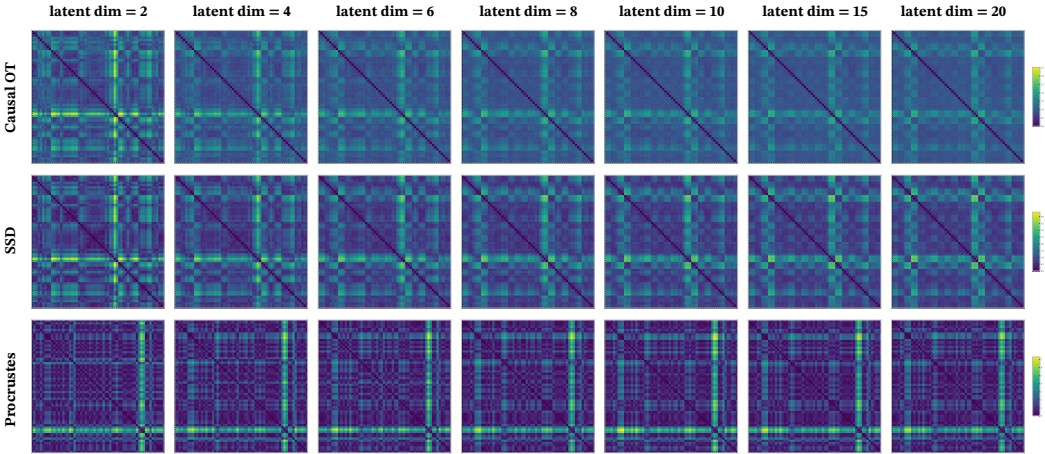

Figure 6: Convergence of normalized shape distances with small number of principal components (PCs). Pairwise distances from three different methods (rows; Causal OT, SSD, Procrustes) between 60 conditional stochastic processes are shown where the latent trajectories are projected onto top $d$ PCs for varying $d \in \{2, 4, 6, 8, 10, 15, 20\}$ (columns). All normalized shape distances converge after $d = 6$.

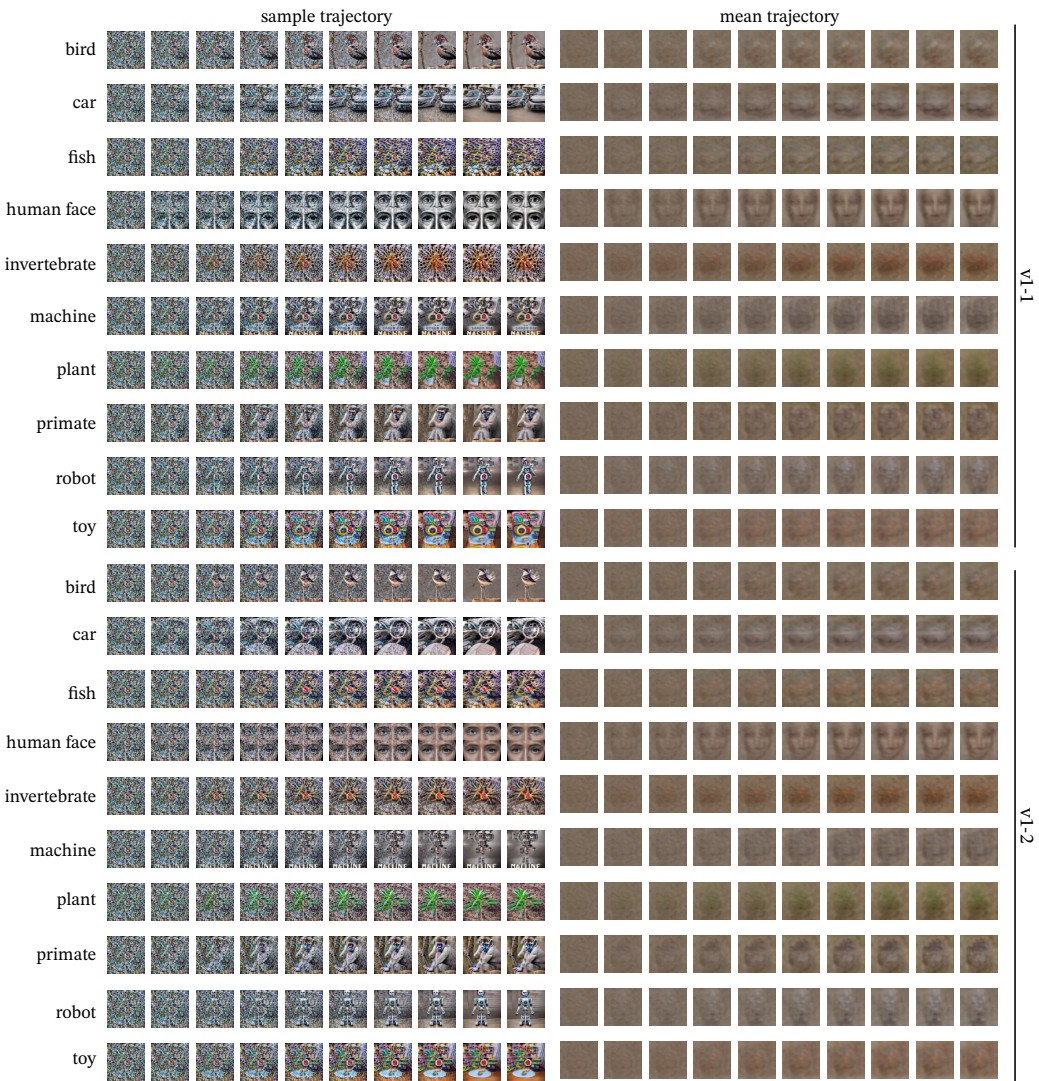

Figure 7: Decoded sample and mean trajectories for all categories in models `v1-1` and `v1-2`.

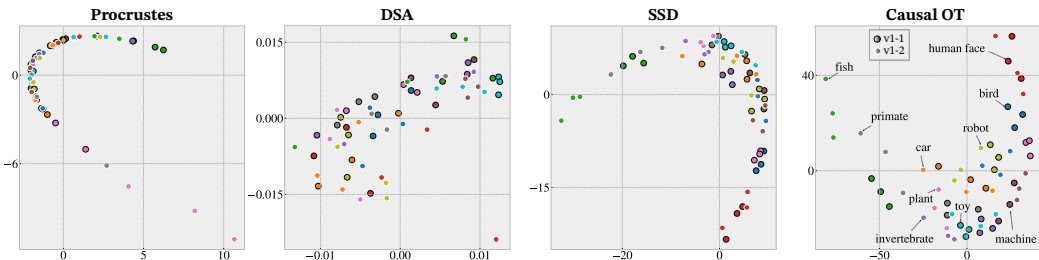

Figure 8: Multi-dimensional scaling of the pairwise shape distances for Procrustes, DSA, SSD, and Causal OT distances corresponding to the distance matrices shown in Fig. 5.

## F COMPARISON OF METRICS ON A SCALAR SYSTEM

Here, using a simple scalar dynamical system model (e.g., the dynamics of a single neuron's responses), we demonstrate that DSA can distinguish systems with varying dynamics (i.e., different underlying vector fields) but the same marginal statistics, while 2-SSD cannot. Conversely, we show that 2-SSD can distinguish systems with varying noise levels but the same dynamics, while DSA cannot. Finally, Causal OT can distinguish the systems in both cases while Procrustes cannot distinguish the systems in either case.

Specifically, consider the scalar autoregressive process (of order 1)

$$x(t) = -ax(t-1) + \delta w(t),$$

where $-1 < a < 1$ is a scalar that determines the autocorrelation of $\{x(t)\}$, $\delta > 0$ is the input noise level and $w = \{w(t)\}$ is a driving white noise process (i.e., a sequence of i.i.d. standard normal random variables). The process $\{x(t)\}$ has a unique stationary distribution given by $\mathcal{N}(0, \sigma^2)$, where $\sigma^2 := \delta^2/(1-a^2)$. Therefore, if the process $\{x(t)\}$ is initialized so that $x(0) \sim \mathcal{N}(0, \sigma^2)$, then its marginal distribution satisfies $x(t) \sim \mathcal{N}(0, \sigma^2)$ for all $t = 1, 2, \ldots$.

### F.1 SYSTEMS WITH DIFFERENT DYNAMICS, SAME MARGINAL STATISTICS

We first show that when the dynamics are different across systems but the marginal distributions are constant across systems, then Causal OT distance and DSA can discriminate between the systems but Procrustes and 2-SSD cannot. For each $a \in \{0.1, 0.3, 0.5, 0.7, 0.9\}$ we let $\delta^2 = 1 - a^2$ so that the stationary distribution of $\{x(t)\}$ is $\mathcal{N}(0, 1)$ provided $x(0) \sim \mathcal{N}(0, 1)$. As $a$ increases, the sample trajectories become smoother and have larger autocovariances (Fig. 9).

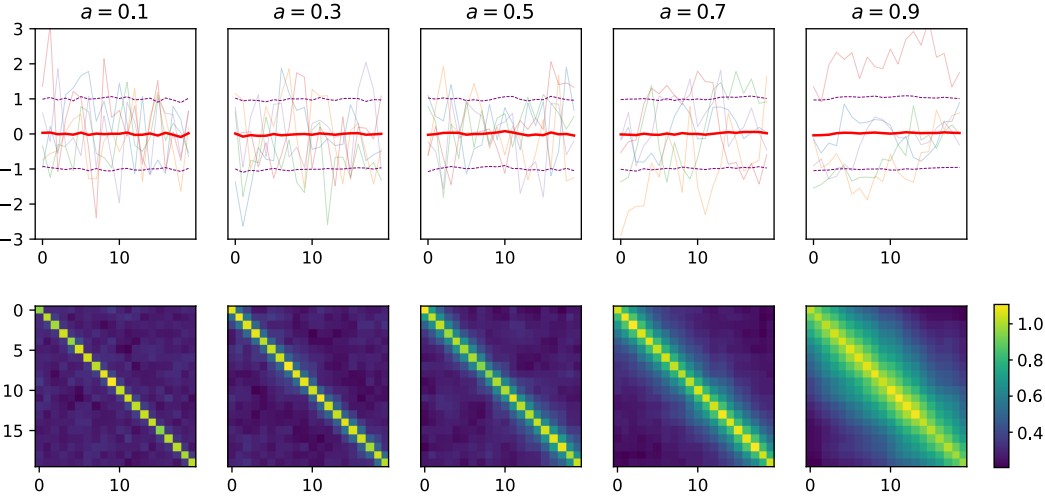

Figure 9: Five autoregressive processes with varying dynamics but constant marginal variances. The top row shows five sample trajectories from each process along with the empirical mean (thick red line) and marginal variance (dashed purple lines). The bottom row shows the autocovariance matrices.

Based on visual inspection of the pairwise distance matrices (Fig. 10), it is clear that Causal OT and DSA differentiate between the systems with varying autocorrelations, but Procrustes and SSD do not.

### F.2 SYSTEMS WITH THE SAME DYNAMICS, DIFFERENT NOISE STATISTICS

Next, we show that when the vector field is fixed across systems but the noise varies, then Causal OT and SSD can discriminate between the systems but Procrustes and DSA cannot. Specifically, we fix the vector field $a = 0.3$ and consider different noise levels $\delta \in \{0.1, 0.2, 0.3, 0.4, 0.5\}$. As

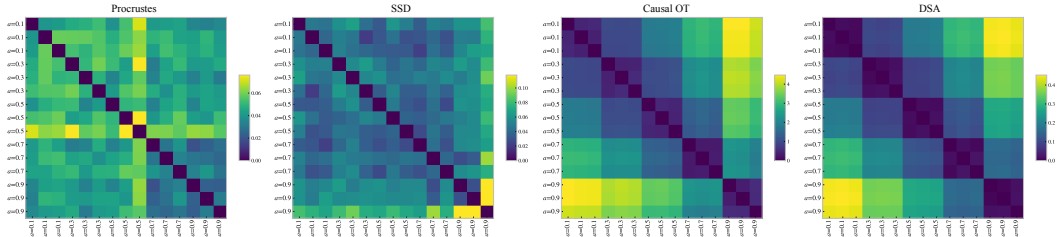

Figure 10: For each $a \in \{0.1, 0.3, 0.5, 0.7, 0.9\}$, we generated 3 sets of 1000 samples trajectories of length $T = 10$, from which we estimated the pairwise distances between the $5 \times 3 = 15$ sets of sample trajectories. To compute the DSA distance we used the following parameters: `n_delays = 3`, `delay_interval = 1`, `rank = 30`.

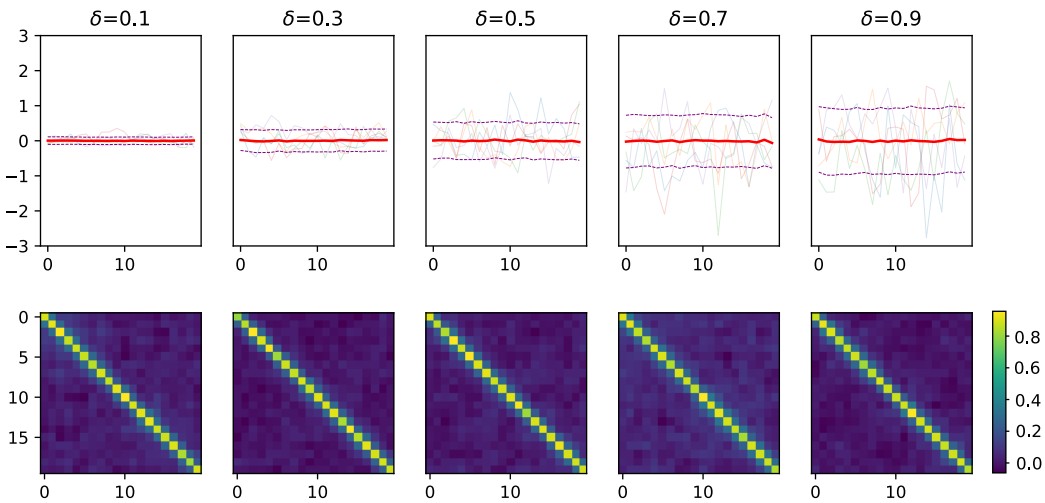

Figure 11: Five autoregressive processes with fixed dynamics and varying noise levels. The top row shows five sample trajectories from each process along with the empirical mean (thick red line) and marginal variance (dashed purple lines). The bottom row shows the autocovariance matrices. To compute the DSA distance we used the following parameters: `n_delays = 3`, `delay_interval = 1`, `rank = 30`.

the noise level increases, the trajectories exhibit larger fluctuations about the mean however, the autocovariance structure remains about constant (Fig. 11).

It is immediately clear from visual inspection that Causal OT and SSD differentiate between the systems with varying noise, but Procrustes and DSA do not (Fig. 12).

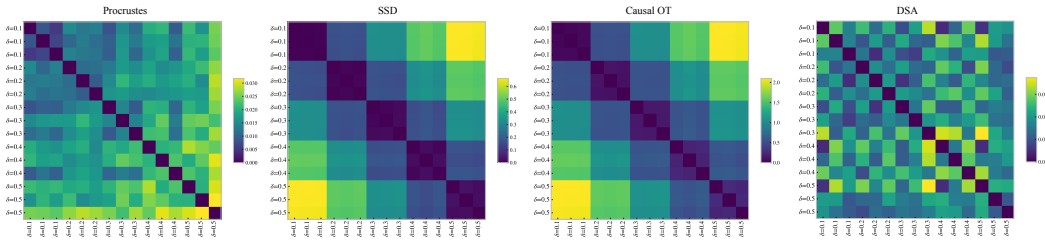

Figure 12: For each noise levels $\delta \in \{0.1, 0.2, 0.3, 0.4, 0.5\}$, we generated 3 sets of 1000 samples trajectories of length $T = 10$, from which we estimated the pairwise distances between the $5 \times 3 = 15$ sets of sample trajectories.

These experiments on a scalar linear systems demonstrate that SSD and DSA can respectively discriminate stochastic and dynamic aspects of stochastic processes; however, on their own they do not capture both elements. On the other hand, Causal OT distance discriminates both stochastic and dynamic characteristics.

# G    SENSITIVITY OF DSA ON HYPERPARAMETERS

DSA (Ostrow et al., 2023) crucially depends on 2 hyperparameters: the number of delays used when creating the Hankel matrix and the rank selected when fitting the reduced-rank regression model to linearly estimate the system's dynamics. We applied DSA (with different choices of delays and rank) to the 2-dimensional dynamical systems from Fig. 2 of the main text to demonstrate the sentivity of the method to the choice of hyperparameters (Fig. 13). In Fig. 2, we presented the result when using the best performing hyperparameters.

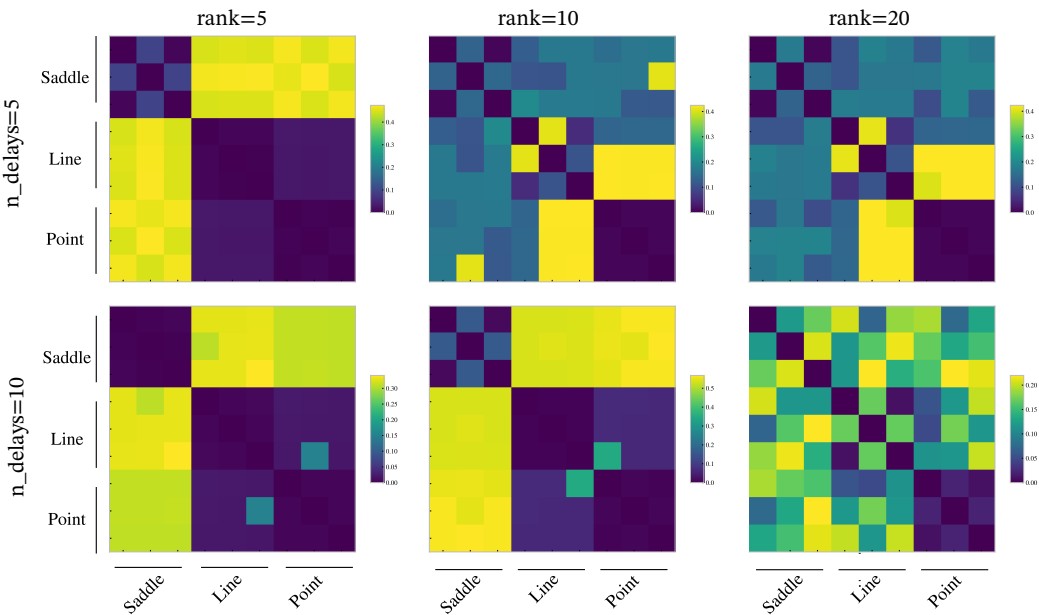

Figure 13: Sensitivity of DSA to hyperparameter selection. We select `n_delays` and `rank` on a grid shown in rows and columns, and run DSA on the same toy dataset as in Fig. 2.

