# OpenReview forum: "Comparing noisy neural population dynamics using optimal transport distances"
_ICLR.cc/2025/Conference — ICLR 2025 Oral_

### Official Review · Reviewer_mGZm · 2024-10-29

**Soundness:** 3
**Presentation:** 4
**Contribution:** 3
**Rating:** 8
**Confidence:** 3

**Summary:**

This paper proposes a new notion of distance between Gaussian processes based on optimal transport between them. The innovation is in considering correlation across time, such that the optimal transport incorporates information across time but respects causality. In particular, this allows the distance to differentiate processes which are indistinguishable when comparing their marginal moments. This distinction is shown to be relevant in a neuroscience-inspired synthetic problem, where distances which compare only marginal moments fail to distinguish small magnitude time segments which have high correlation with other time segments. These distances are also used to compare the output trajectories generated by denoising diffusion image models.

**Strengths:**

* The presentation throughout is very clear and a pleasure to read.
* The motivation of the method through prior work is exceptionally good.
* The method this paper proposes is intuitive but original, fairly simple, well-motivated, and seems to be a substantial extension of previous work.
* The experimental evaluation is appropriate.

**Weaknesses:**

* The experimental evaluation is a bit terse, and doesn't demonstrate a clear advantage for this paper's method over SSD across the board. The method is only evaluated on synthetic neural data even though neuroscience is the clearest application for these ideas. Why not try the it on the actual reaching data which inspired the synthetic dataset?
* I'm unsure how important the problem this method addresses is -- the extent to which it's useful to compute distances between stochastic processes for applications like comparing diffusion models or brains is not clear.

**Questions:**

* I would appreciate some discussion of the derivation of the causal OT distance in the main body. This seems important to justifying how it can be viewed as an OT distance.

---

> ### Author Response · Authors · 2024-11-21
>
> Thank for your thoughtful review. We are very pleased that you found the paper a pleasure to read!
>
> - We plan to apply our method to neural data and are actively doing so in ongoing work (see our general response **Algorithmic complexity and evaluation on neural data**).
> - We are working on including the important aspects of the derivation while staying within the 10 page limit.
> - We will improve the introduction to make the motivation more clear. We give a brief review of relevant literature below.
>   - Artificial deep networks perform computations by cascaded manipulations of vector-valued input vectors (“representations”) and this is thought to be a reasonable approximation to what happens in biological systems. The geometry of these hidden layer representations ought to give us hints as to how the systems work as well as their vulnerabilities (e.g. Kriegeskorte & Kievit, 2013; Trends in cognitive sciences), and a large body of work leveraged representational similarity measures like centered kernel alignment (CKA; Kornblith et al. 2019), canonical correlations analysis (CCA; Raghu et al. 2017), and Procrustes distance (Williams et al., 2021) to compare the geometry of neural representations across different artificial and biological systems. These distance measures can be used for a variety of things like comparing whether deep vs. wide neural networks utilize similar representations (Nguyen et al., 2020). Another example is the recently proposed “Platonic Representation Hypothesis” (Huh et al., 2024), which suggests that deep networks converge onto the same representation when trained on very large datasets. Quantifying this hypothesis will require distance metrics between systems.
>   - Our paper extends this body of work to handle stochastic and dynamic representations. Duong et al. (2023) recently proposed a metric for stochastic, non-dynamic neural representations, and Ostrow et al. (2023) recently proposed a metric for dynamic, non-stochastic representations. Both were published at high quality venues, ICLR and NeurIPS, and are well cited. Ours is an important step forward because it leverages both dynamic and stochastic structure to form a distance measure. Figure 1 in our paper aims to illustrate the advantages of this.

---

> > ### Comment · Reviewer_mGZm · 2024-11-25
> >
> > Thanks for the thoughtful response. I maintain that this is a good paper and stand by my original score.

---

### Official Review · Reviewer_Sz7A · 2024-10-30

**Soundness:** 3
**Presentation:** 4
**Contribution:** 3
**Rating:** 8
**Confidence:** 4

**Summary:**

This paper introduces a new metric for comparing noisy neural trajectories. The metric is based on computing optimal transport distances between gaussian processes. While the metric can in principle be used to compare any two stochastic processes, the authors justify the metric rigorously for linear-time varying stochastic processes. To support their theoretical claims, the authors provide several numerical examples where their new metric performs better than two previously proposed metrics (Procrustes and SSD).

**Strengths:**

The paper is _very_ well-written, well-motivated, and timely. The mathematical exposition is especially clear. The numerical experiments are also nice.

**Weaknesses:**

The major weakness of the paper is the lack of numerical comparisons to Dynamical Similarity Analysis (DSA), introduced by Ostrow et al, 2023. This omission is very puzzling, since the authors mention DSA early and often, and even use some of the same tasks as DSA (Fig 2).

**Questions:**

Why not compare to DSA? Even if DSA does "better" on some of the tasks, that does not make your method less valuable. As you point out, DSA is only theoretically justified for deterministic systems. Therefore, _I will be happy to raise my score if these comparisons are included, regardless of comparative performance._

EDIT 11/21--The authors have included the analysis I asked for. I have accordingly raised my score.

### Minor points:
- L046-L047: The phrasing makes it sound like both studies addressed stochasticity and dynamics.
- L050: You don't actually show this for DSA, as far as I can tell.
- Equation 3: There is a missing time dependence in $A^\top$.

---

> ### Author Response · Authors · 2024-11-21
>
> Thank you for your review. We are very pleased to hear that you found that paper well-written, well-motivated at timely.
>
> We agree that including a comparison with DSA is important and have added these comparisons (see our general response).

---

### Official Review · Reviewer_4ecC · 2024-11-05

**Soundness:** 3
**Presentation:** 3
**Contribution:** 4
**Rating:** 8
**Confidence:** 4

**Summary:**

This paper presents a novel metric for comparing noisy neural population dynamics based on the notion of causal optimal transport, which respects temporal causality. Other methods in the field generally assume deterministic or static neural activity, missing key aspects of biological and artificial neural systems. They validate their metric in a biological task (synthetic motor control experiment) and artificial task (conditional image generation in diffusion) and find that their metric distinguishes different neural trajectories properly.

**Strengths:**

- The paper is well written, providing a good motivation and background on the limitations of current metrics such as SSD and Procrustes Shape Analysis in distinguishing different types of neural dynamics. The mathematics of optimal transport is well presented and motivated with good explanations of the intuition behind the new metric.
- The paper provided three sets of experiments: scalar task comparing the different metrics mathematically, synthetic for biological motor control, and latent diffusion models.
- The discussion on disentangling recurrent dynamics from input-driven dynamics really highlights importance of considering temporal dependencies in the data.

**Weaknesses:**

- As mentioned by the authors themselves, the metric lies on an assumption of Gaussian neural processes, which does not hold for biological and artificial neural systems. It may not apply to non-Gaussian data as it won't be able to capture the higher-order statistical dependencies between the trajectories.
- Estimating full covariance matrices for high-dimensional neural data (large N and T) is computationally intensive and probably requires a prohibitive number of samples.
- The paper mentions Dynamical Systems Analysis by Ostrow et al. in the end of the paper as a possible future direction. It would be nice if the authors can actually include a comparison to that method in the current paper, as they compare Causal OT to Procrustes and SSD, but DSA seems to be a better method for studying neural dynamics than both of them.

**Questions:**

- I would love to see theoretical or empirical analysis of the metric's robustness to deviations from Gaussianity. I know this was briefly mentioned in the section on Diffusion, but I would like to see a demonstration of the metric's performance as you increase the impact of higher order statistical moments such as skewness or kurtosis.
- The metric is computational expensive. Can it be reduced using some dimensionality reduction techniques that preserves stochastic and temporal structure (maybe Dynamic Mode Decomposition or Tensor Factorization)? Can you impose structure on the covariance matrices, such as low-rank approximations or sparsity constraints, to reduce the number of parameters to estimate?
- Can you comment on how this method is affected by the observations in Qian et al. 2024 (Partial observation can induce mechanistic mismatches in data-constrained models of neural dynamics) that points out challenges in identifying mechanisms in neural data (such as line attractors) based on partial observation of neural data.
- I would like to see experiments comparing Causal OT to DSA in some way.
- I would like to see an example applies to actual neural data such as calcium imaging or electrophysiological recordings available publicly and see if the metric can distinguish between neural trajectories coming from different experimental conditions. This might be hard to do within the review period, but it would greatly improve the paper's quality and would motivate me to increase the score substantially.

---

> ### Author Response · Authors · 2024-11-21
>
> Thank for your careful reading of our paper and for your thoughtful comments. We have addressed your concerns about the Gaussian process assumption, the computational inefficiency of our algorithm and the comparison to DSA in our general response.
>
> In response to your questions:
> - Importantly, our metric only depends on first- and second-order statistics of processes ${\bf x}$ and ${\bf y}$. Therefore, increasing the deviation in the higher-order moments between the processes ${\bf x}$ and ${\bf y}$ will not affect the Causal OT distance between ${\bf x}$ and ${\bf y}$ as long as their first- and second-order moments remain the same. We agree it would be interesting to develop methods that can capture differences in higher-order moments. One possibility (following Duong et al. ICLR, 2023) is to extend energy distances to compare stochastic processes.
> - Indeed, this is an important point: our method is computationally intensive when applied to high-dimensional neural data. We are actively thinking about methods to improve the efficiency of our method when applied to high-dimensional neural data (see our general response **Algorithmic complexity and evaluation on neural data**).
> - Our method does not make a low-dimensional assumption, so in general it will not suffer from the problems documented in Qian et al. 2024. We agree it would be interesting to test the ability of our method to differentiate line attractors versus feedforward amplification via non-normal dynamics.
> - See our general response **Comparison with DSA**
> - We are intent on applying our method to neural data. We view this as a methodological paper and plan to include neural data in follow up work (see our general response **Algorithmic complexity and evaluation on neural data**).

---

> ### Author Response · Authors · 2024-11-24
>
> Dear Reviewer,
>
> Thank you again for taking the time to provide feedback on our submission. We have responded to your concerns in our rebuttal and would greatly appreciate it if you could review our clarifications and let us know if there are any further points we can address before the discussion period ends. Thank you!

---

> > ### Comment · Reviewer_4ecC · 2024-12-02
> > **Increase in Score**
> >
> > Thank you for the explanations, and especially the additional comparisons to DSA. I think these experiments definitely add value in demonstrating the advantage of this metric. I would love to see you incorporate these extensions you propose in your response into future work, as this seems like an exciting avenue for studying neural activity.

---

### Author Response · Authors · 2024-11-21
**General response (summary of changes)**

Thank you for your careful reviews and thoughtful comments. We’re glad that you found the paper well written and well motivated. We sincerely appreciate your constructive critiques and have revised our paper accordingly. We list the changes we made to the manuscript and address broad issues here, and respond individually to each review below.

# Major additions

1. We now include comparisons to DSA in all of the numerical experiments (Figs. 2, 4, 5).
2. We added Appendix F which compares all of the methods (Procrustes, SSD, DSA and Causal OT) on a scalar autoregressive (AR) process.
3. We added Appendix G which tests the sensitivity of DSA to the following hyperparameters: **number of delays** used to construct the Hankel matrix and **rank** used when solving the reduced-rank regression problem.

# Minor changes

1. We moved the discussion about “recurrent dynamics” versus “feedforward drive” earlier in section 2 because it’s one of the main motivations for developing the metric.
2. We slightly generalized the distance to depend on a parameter $0\le\alpha\le2$, which adjusts the relative weight that is placed on the means versus the covariances. When $\alpha=0$, the distance reduces to Procrustes distance, whereas when $\alpha=2$ the distance only depends on the covariance structure.
3. Notation: we now use ${\bf x}$ and ${\bf y}$ instead of ${\bf x}^A$ and ${\bf x}^B$ to denote two *processes*. This is to avoid confusion with the *matrices* ${\bf A}$ and ${\bf B}$, which were not always related to the processes.

---

> ### Author Response · Authors · 2024-11-21
> **Broad issues**
>
> # Comparison with DSA
>
> Reviewers **4ecC** and **Sz7A** state that a comparison of Causal OT with DSA would strengthen our results. We now provide comparisons of our method with DSA on all of the experiments that we performed (except for the 2-step example since DSA requires delay-embedding the process). We also added 2 appendices that further explore the performance of DSA and how it compares with Causal OT (as well as with Procrustes and SSD). While our results are not comprehensive, we generally find the following hold:
> - DSA does not distinguish between systems with the same underlying dynamics but different noise statistics (Appendix F). This is not surprising given that DSA was not designed to differentiate between systems with different noise statistics.
> - DSA effectively distinguishes between systems in some examples (Figs. 2 and 4) but not all examples (Fig. 5), and the efficacy of DSA is sensitive to the choice of hyper-parameters (Appendix G).
>
> Here, we provide a more detailed account of what we added:
> - We include a description of DSA (section 2.3).
> - We include a comparison with DSA in Figs. 2, 4 and 5.
>   - Fig. 2: DSA distinguishes between all three systems. However, this required carefully choosing the hyperparameters (number of delays and rank). For other choices of hyperparameters, DSA fails to distinguish between the systems (Appendix G).
>   - Fig 4: DSA distinguishes between the preparatory activity and the readout activity, even when preparatory activity is low. This is likely because changing the levels of preparatory activity does not change the topology of the system.
>   - Fig. 5: DSA fails to distinguish between different conditional processes. Due to time constraints, we did not perform an exhaustive test of hyperparameter choices for DSA, so we cannot rule out the possibility that DSA would effectively distinguish the conditional processes using another choice of hyperparameters.
> - We compare Procrustes, SSD, Causal OT and DSA on a scalar AR process with varying dynamics and noise levels (Appendix F).
> Procrustes fails to distinguish any of the systems (as expected since the systems are all mean zero).
> SSD distinguishes systems with the same dynamics but different noise levels; however, SSD does not distinguish systems with the same marginal noise statistics but different dynamics.
> DSA distinguishes systems with the same marginal noise statistics but different dynamics; however, DSA does not distinguish systems with the same dynamics but different noise levels.
> Causal OT distinguishes all of the systems.
>
> # Algorithmic complexity and evaluation on neural data
>
> Reviewer **4ecC** expressed concern about the computational complexity of the proposed method and Reviewers **4ecC** and **mGZm** stated that they would like to see the method tested on neural data.
> 1. While the focus of this work has been laying the foundation for developing distances that respect stochasticity and dynamics in data, it is indeed critical to have algorithms for efficient computation of these distances. We discuss this in the Limitations section of the paper, and reference two related works that can enable efficient computation of Causal OT distance. Importantly, one of the two papers (Nejatbakhsh et al. NeurIPS 2023) explicitly models marginal covariances without incorporating across time covariances, while the other work (Duncker et al. ICML 2019) models the low-dimensional dynamics through latent variables with assumptions on the covariances.
> 2. In ongoing work, we’re combining these two techniques and developing a latent variable model that is flexible enough to capture dynamics and both marginal and across time covariances, which can then be used to estimate distances between neural processes. The generative model and inference are involved and require more space for exposition. We have applied the model to multiple neural datasets and estimated covariances; however, this process involves close collaboration with experimental neuroscientists and we believe this is beyond the scope of this methodological paper.

---

> ### Author Response · Authors · 2024-11-21
> **Broad issues continued**
>
> # Gaussian process (GP) assumption
>
> Reviewer **4ecC** raised concerns about the assumption of Gaussianity, which does not hold for neural processes. We would like to make a few clarifications.
> - While it is true that the GP is the running example in the paper, it’s worth clarifying that Causal OT is a pseudometric over *all stochastic processes* in the sense that $d({\bf x},{\bf y}) = 0$ if and only if the mean and covariance of ${\bf x}$ and ${\bf y}$ are equal. This follows easily from the Bures + adapted Bures distances being a proper metric on covariance. Thus, strictly speaking, none of the distances we compute assume a GP.
> - If we do make a GP assumption, there are some additional nice benefits. The distances we propose are proper metrics over Gaussian processes in that if ${\bf x}$ and ${\bf y}$ are GPs then $d({\bf x},{\bf y}) = 0$ if and only if ${\bf x} ={\bf y}$. Additionally, the interpretation of the distance in terms of optimal transport holds only when ${\bf x}$ and ${\bf y}$ are GPs. But these are just nice bonus features.
> - Causal OT distances that incorporate higher order moments are defined for arbitrary (GP or non-GP) stochastic processes (Backhoff et al. SIAM J. Opt., 2017). However, statistical estimators for these quantities suffer a severe curse of dimensionality and are generally viewed as intractable (e.g. Boissard & Le Gouic AIHP, 2014) .
> - We agree with the reviewer that the distance we present in the paper (Causal OT on GPs) can fail to distinguish adversarial cases where the mean and covariance of the processes are tuned to be the same while higher order moments are different. However, despite their limitations, GPs are a popular framework for modeling neural dynamics (e.g. Duncker et al. ICML 2019).

---

### Meta-Review · Area_Chair_cfiD · 2024-12-18

**Metareview:**

The authors present a novel method for estimating the similarity of two sets of neural data using causal optimal transport. While there was some concern about the lack of comparison on real experimental data, the authors agreed the method was novel and useful and the paper was well written, and there was a clear consensus that the paper should be accepted. I agree.

**Additional Comments On Reviewer Discussion:**

The reviewers all engaged constructively with the authors.

---

### Decision · Program_Chairs · 2025-01-22

Accept (Oral)